# Topological invariant and anomalous edge modes of strongly nonlinear systems

Di Zhou [1✉], D. Zeb Rocklin[2], Michael Leamy [3] & Yugui Yao [1✉]

Despite the extensive studies of topological states, their characterization in strongly nonlinear classical systems has been lacking. In this work, we identify the proper definition of Berry phase for nonlinear bulk waves and characterize topological phases in one-dimensional (1D) generalized nonlinear Schrödinger equations in the strongly nonlinear regime, where the general nonlinearities are beyond Kerr-like interactions. Without utilizing linear analysis, we develop an analytic strategy to demonstrate the quantization of nonlinear Berry phase due to reflection symmetry. Mode amplitude itself plays a key role in nonlinear modes and controls topological phase transitions. We then show bulk-boundary correspondence by identifying the associated nonlinear topological edge modes. Interestingly, anomalous topological modes decay away from lattice boundaries to plateaus governed by fixed points of nonlinearities. Our work opens the door to the rich physics between topological phases of matter and nonlinear dynamics.

[1] Key Lab of Advanced Optoelectronic Quantum Architecture and Measurement (MOE) and School of Physics, Beijing Institute of Technology, Beijing 100081, China. [2] School of Physics, Georgia Institute of Technology, Atlanta, GA 30332, USA. [3] School of Mechanical Engineering, Georgia Institute of Technology, Atlanta, GA 30332, USA. ✉email: dizhou@bit.edu.cn; ygyao@bit.edu.cn

The advent of topological band theory has led to the burgeoning field of "topological phases of matter" which manifest exotic properties, such as surface conduction of electronic states, and wave propagation insensitive to backscattering and disorder[1–4]. In classical structures[5–12], enormous efforts have been devoted to topological states that emulate their quantum analogs and enable many pioneering applications[7,13–28]. However, most studies of classical structures are limited to linear topological band theory, whereas nonlinear topological systems are not fully understood yet.

Nonlinear dynamics are more ubiquitous in nature, such as electrical circuits composed of nonlinear elements[7,29], nonlinear elastic and mechanical structures[25,30–34], nonlinear origami systems[35], evolutionary dynamics of biological cycles[36,37], second-harmonic generation of optic materials[38,39], and cold atoms in optical lattices[40,41]. To date, few studies are addressed in nonlinear topological photonics[14,42–45] and Bose-Einstein condensates[46,47], but the interactions are limited to Kerr-like nonlinearities controlled by field intensities. These Kerr-like nonlinearities are the simplest ones that grant sinusoidal nonlinear bulk waves and thus the topological invariants[45,48] are the same as those in linear theories.

However, the majority of classical structures are beyond Kerr-like nonlinearities, such as the aforementioned electrical, mechanical, biological, and optic systems. Nonlinear bulk modes cannot be solved analytically[37,49], leading to topological invariants in these strongly nonlinear systems undefined. Though boundary modes remain topologically protected in the weakly nonlinear regime[7,26,34], strong nonlinearities may destroy their topological nature by breaking the intrinsic symmetries[46,47], and existing linear and weakly nonlinear topological theories are not always correct to predict their strongly nonlinear topological properties[35]. Moreover, it is intriguing to ask what exotic physics and unconventional attributes arise when topology meets universal strong nonlinearities. Thus, it is demanding to invoke the topological number that precisely describes the topological attributes of "beyond-Kerr" strongly nonlinear systems.

This work investigates the topological invariant and properties of 1D generalized nonlinear Schrödinger equations beyond Kerr-like nonlinearities. In spite of the remarkable different physical origins of mechanical isostatic structures[35], electrical circuits[7,29], deep water waves[50], and bio-physical cycles[36,51–53], their dynamics are commonly described by generalized nonlinear Schrödinger equations, which we adopt to study theoretically, for future nonlinear topological experiments. The nonlinear parts of interactions are comparable to the linear ones and perturbation theory breaks down, which we designate the "strongly nonlinear regime". We limit our considerations within the amplitude range[32,33] that chaos does not occur. Consequently, nonlinear bulk modes[31,54] are remarkably distinct from sinusoidal waves (e.g., Fig. 1b and SI. Fig. 11c). We develop the proper definition of Berry phase of nonlinear bulk modes. By adopting a symmetry-based analytic treatment, we demonstrate the quantization of Berry phase in reflection-symmetric systems, regardless of the availability of linear analysis. The emergence of nonlinear topological edge modes is associated with a quantized Berry phase that protects them from disorders. Interestingly, exotic boundary responses arise when topology meets nonlinearity. Instead of exponentially localizing on lattice boundaries, topological edge modes exhibit anomalous behaviors that decay to a plateau governed by the stable fixed points of nonlinearities.

## Results

**Quantized Berry phase of nonlinear bulk modes:** Generalized nonlinear Schrödinger equations are widely studied in classical

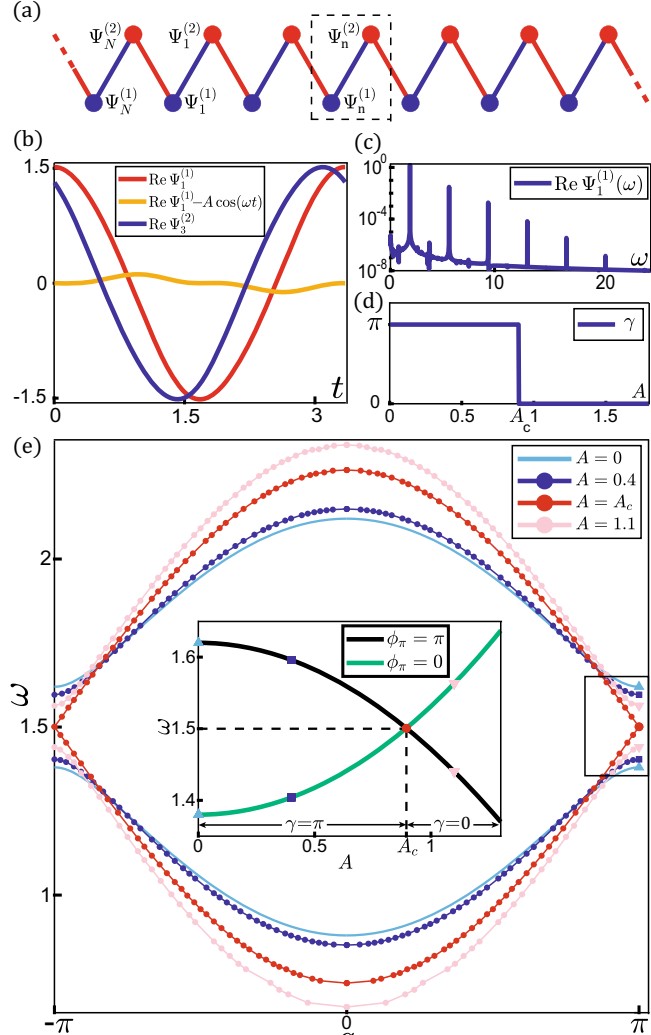

**Fig. 1 The minimal model of nonlinear SSH chain. a** Schematic illustration of the lattice subjected to PBC. Unit cell is enclosed by black dashed box. Red and blue bonds represent intra-cell and inter-cell couplings. **b** A nonlinear bulk mode computed by shooting method[62–64] with amplitude $A = 1.5$ and wave number $q = 4\pi/5$. Red and blue curves are the wave functions of $n = 1$ and 3 sites, respectively. Orange curve shows the noticeable difference between nonlinear mode and sinusoidal function. **c** Frequency profile of nonlinear bulk mode in (**b**). **d** Numerical computation of the amplitude dependence of nonlinear Berry phase in Eq. (3). The algorithmic details are elaborated in SI. II.C. (**e**), nonlinear band structures $\omega = \omega(q, A)$ plotted for bulk mode amplitudes from $A = 0$ to 1.1. The red curves touch for the topological transition amplitude $A_c = 0.8944$ at $\omega = \epsilon_0 = 1.5$. The inset elaborates on the gap-closing transition amplitude $A_c$ at which band inversion occurs.

systems like mechanically isostatic lattices[35], electrical circuits[7,29], deep water waves[50], and nonlinear Markov networks of biochemical dynamics[36,51–53]. Their equations of motion are summarized as the general form in Eq. (1) below. We study nonlinear bulk modes, from which we define Berry phase and demonstrate its quantization in reflection-symmetric models. In Sec.IV, we propose an electrical circuit as one of the classical systems that yield Eq. (1).

The considered model is a nonlinear SSH[55] chain composed of $N$ classical dimer fields $\Psi_n = (\Psi_n^{(1)}, \Psi_n^{(2)})^\top$ ($\top$ is matrix transpose) coupled by nonlinear interactions, as represented pictorially in Fig. 1a. The chain dynamics is governed by the 1D generalized

nonlinear Schrödinger equations,

$$i\partial_t \Psi_n^{(1)} = \epsilon_0 \Psi_n^{(1)} + f_1\left(\Psi_n^{(1)}, \Psi_n^{(2)}\right) + f_2\left(\Psi_n^{(1)}, \Psi_{n-1}^{(2)}\right),$$
$$i\partial_t \Psi_n^{(2)} = \epsilon_0 \Psi_n^{(2)} + f_1\left(\Psi_n^{(2)}, \Psi_n^{(1)}\right) + f_2\left(\Psi_n^{(2)}, \Psi_{n+1}^{(1)}\right),$$

(1)

subjected to periodic boundary condition (PBC), where $\epsilon_0 \geq 0$ is the on-site potential, and $f_i(x, y)$ for $i = 1$ and $i = 2$ stand for intracell and intercell nonlinear couplings, respectively. $f_i(x, y)$ are real-coefficient general polynomials of $x$, $x^*$, $y$, and $y^*$ ($*$ represents complex conjugate), which offer time-reversal symmetry[3]. Given a nonlinear solution $\Psi_n(t)$, time-reversal symmetry demands a partner solution $\Psi_n^*(-t)$, as demonstrated in SI. II.A. For systems such as those with Bose-Einstein condensates[56], $|\Psi(\vec{r}, t)|^2$ corresponds to a particle number density and third-order nonlinearities are thus limited to Kerr interactions $|\Psi|^2 \Psi$ to enforce particle number conservation; in our case the fields do not correspond to particle densities and more general nonlinearities beyond Kerr-like interactions are thus permitted.

In linear regime, the polynomials are approximated as $f_i(x, y) \approx c_i y$ ($c_{i=1,2} > 0$) to have "gapped" two-band models when $c_1 \neq c_2$. The bulk mode eigenfunctions are sinusoidal in time, and Berry phase is quantized by reflection symmetry. In the "strongly nonlinear regime" where nonlinear interactions become comparable to the linear ones, nonlinear bulk modes are significantly different from sinusoidal waves (e.g. Fig. 1b, SI. Fig. 11c), and the frequencies naturally deviate from their linear counterparts. The nonlinearities become increasingly important as the bulk mode amplitude rises. Hence, the frequency of a nonlinear bulk mode is controlled both by wave number and amplitude. We thus define nonlinear band structure[7,48] $\omega = \omega(q \in [-\pi, \pi], A)$ as the frequencies of nonlinear bulk modes for given amplitude $A$. We consider the simple case that nonlinear bulk modes are always non-degenerate (i.e., different modes at the same wave number have different frequencies) unless they reach the topological transition amplitude when the nonlinear bands merge at the band-touching frequency. Hence, given the amplitude, frequency, and wave number, a nonlinear bulk mode is uniquely defined. Extended from gapped linear models, the lattice is a "gapped two-band nonlinear model". In what follows, we define Berry phase for nonlinear bulk modes of the upper-band by adiabatically evolving the wave number across the Brillouin zone.

The considered nonlinear bulk mode is spatial-temporal periodic. It takes the plane-wave nonlinear normal modes in translationally invariant systems[31,57–61] (also dubbed as "nonlinear plane waves"),

$$\Psi_q = \left(\Psi_q^{(1)}(\omega t - qn), \Psi_q^{(2)}(\omega t - qn + \phi_q)\right)^\top,$$

(2)

where $\omega$ and $q$ are the frequency and wave number, respectively. $\Psi_q^{(j=1,2)}(\theta)$ are $2\pi$-periodic wave components, where the phase conditions are chosen by asking $\mathrm{Re}\,\Psi_q^{(j)}(\theta = 0) = A$, and $A \stackrel{\text{def}}{=} \max(\mathrm{Re}\,\Psi_q^{(j)})$ is the amplitude. This is analogous to the phase condition $\mathrm{Re}\,\Psi(t = 0) = \max(\mathrm{Re}\,\Psi(t))$ adopted in Schrödinger equation in order to have the eigenfunctions $\Psi(t) = |\Psi| e^{-i\epsilon t/\hbar}$. Following this condition, $\phi_q$ in Eq. (2) characterizes the relative phase between the two wave components. Nonlinear bulk modes are not sinusoidal. They fulfill $i\partial_t \Psi_q = H(\Psi_q)$, where $H(\Psi_q)$ is the nonlinear function determined by Eq. (1) and is elaborated in SI. I. Given the band index and the amplitude $A$ of a nonlinear bulk mode, we find that $\omega$, $\phi_q$, and the waveform are determined by the wave number $q$. Equation (2) is solved by the numerical shooting method[62–64] that applies for general nonlinearities, as detailed in SI. III.B.

The ansatz in Eq. (2) correctly states the periodicity of nonlinear bulk modes based on the following reasons. First, existing

works[58] manifest this form of nonlinear plane waves, such as classical Minkowskian Yang-Mills theory[59], compressible atmosphere[60], porous media[61], and mechanical lattices[32,54]. Second, typical studies on weakly nonlinear bulk modes[11,25,31–33,54,65] reveal that the dynamics of all high-order harmonics are controlled by the single variable $\theta = \omega t - qn$: $\Psi_q^{(j)} = \sum_l \psi_{l,q}^{(j)} e^{-il(\omega t - qn)}$, where $\psi_{l,q}^{(j)} = (2\pi)^{-1} \int_0^{2\pi} e^{il\theta} \Psi_q^{(j)} d\theta$ is the $l$-th Fourier component of $\Psi_q^{(j)}$. Third, numerical experiments such as shooting method (see Fig. 1b, SI. Fig. 11a, c, and Refs. [62–64]) manifest non-dispersive, plane-wave like bulk modes in the strongly nonlinear regime. Finally, it is demonstrated in SI. III.C that the analytic solutions of nonlinear bulk modes at high-symmetry wave numbers are in perfect agreement with Eq. (2). Consequently, the frequencies and band structure of temporal-periodic nonlinear bulk waves[32,54,58–61] are characterized by the wave number $q$ as well, as pictorially indicated in Fig. 1e.

While the ansatz in Eq. (2) captures the periodicity of nonlinear bulk states, it cannot describe temporal-periodic nonlinear modes with spatially inhomogeneous amplitudes, such as soliton excitations[32,63] and nonlinear localized modes[31]. Corresponding detailed discussions are addressed in SI. I.

We realize the adiabatic evolution of wave number $q(t)$ traversing the Brillouin zone from $q(0) = q$ to $q(t) = q + 2\pi$, while the amplitude $A$ remains unchanged during this process. According to the nonlinear extension of the adiabatic theorem[23,24,66,67], a system $H(\Psi_q)$ initially in one of the nonlinear modes $\Psi_q$ will stay as an instantaneous nonlinear mode of $H(\Psi_{q(t)})$ throughout this procedure, provided that the nonlinear mode $\Psi_q$ is stable[67] within the amplitude scope of this paper. Due to the symmetry constraints of the nonlinear motion equations, we demonstrate that all nonlinear bulk states are marginally stable within Floquet analysis[64,68,69] (see SI. II.E for details). Mode stability is further confirmed in SI. III.B via the algorithm of self-oscillation[11,25,54]. Therefore, the only degree of freedom is the phase of mode. At time $t$, the mode is $\Psi_{q(t)}(\int_0^t \omega(t', q(t')) dt' - \gamma(t))$, where $\gamma(t)$ defines the phase shift of the nonlinear bulk mode in the adiabatic evolution. The dynamics of $\gamma$ is depicted by $(d\gamma/dt)(\partial\Psi_q/\partial\theta) = (dq/dt)(\partial\Psi_q/\partial q)$. After $q$ traverses the Brillouin zone, the wave function acquires an extra phase $\gamma$ dubbed Berry phase of nonlinear bulk modes,

$$\gamma = \oint_{\mathrm{BZ}} dq \frac{\sum\limits_{l \in \mathcal{Z}} \left( l \left|\psi_{l,q}^{(2)}\right|^2 \frac{\partial\phi_q}{\partial q} + i \sum\limits_j \psi_{l,q}^{(j)*} \frac{\partial\psi_{l,q}^{(j)}}{\partial q} \right)}{\sum\limits_{l' \in \mathcal{Z}} l' \left( \sum\limits_{j'} \left|\psi_{l',q}^{(j')}\right|^2 \right)},$$

(3)

where $j, j' = 1, 2$ denote the two wave components, and the mathematical derivations are displayed in SI. I. In general, $\gamma$ is not quantized unless additional symmetry properties are imposed on the model, which we will discuss below. We note that the eigenmodes of linear problems as well as Kerr-like nonlinear systems[45,48] are sinusoidal in time, which reduces Eq. (3) to the conventional form[66] $\gamma_{linear} = \oint_{\mathrm{BZ}} dq\, i\langle\Psi_q|\partial_q|\Psi_q\rangle$.

Now we demonstrate that Berry phase defined in Eq. (3) is quantized by reflection symmetry. The model in Eq. (1) respects reflection symmetry, which means that the nonlinear equations of motion are invariant under reflection transformation,

$$\left(\Psi_n^{(1)}, \Psi_n^{(2)}\right) \rightarrow \left(\Psi_{-n}^{(2)}, \Psi_{-n}^{(1)}\right).$$

(4)

Given a nonlinear bulk mode $\Psi_q$ in Eq. (2), reflection transformation demands a partner solution $\Psi_{-q}' = (\Psi_q^{(2)}(\omega t + qn), \Psi_q^{(1)}(\omega t + qn - \phi_q))^\top$ that also satisfies the model. On the other hand, a nonlinear bulk mode of wave number $-q$ is by definition denoted as $\Psi_{-q} = (\Psi_{-q}^{(1)}(\omega t + qn), \Psi_{-q}^{(2)}(\omega t + qn + \phi_{-q}))^\top$. Since there is no degeneracy of nonlinear bulk modes, $\Psi_{-q}'$ and $\Psi_{-q}$ have

to be identical, which imposes the constraints

$$\phi_{-q} = -\phi_q \bmod 2\pi, \text{ and } \Psi_q^{(2)} = \Psi_{-q}^{(1)}. \quad (5)$$

Thus, the Fourier components of nonlinear bulk modes satisfy $\psi_{l,q}^{(2)} = \psi_{l,-q}^{(1)}$. This relationship, together with Eq. (5), is the key to quantize the Berry phase in Eq. (3) (details in SI. II.B),

$$\gamma = \tfrac{1}{2} \oint_{BZ} \frac{d\phi_q}{dq} dq = \phi_\pi - \phi_0 = 0 \text{ or } \pi \bmod 2\pi, \quad (6)$$

where $\phi_{q=0,\pi}$ are the relative phases of the upper-band nonlinear modes at high-symmetry points. They are determined by comparing the frequencies $\omega(\phi_q = 0)$ and $\omega(\phi_q = \pi)$ for $q = 0$ and $\pi$. $\gamma = \pi$ if $\omega(\phi_0 = 0)$ and $\omega(\phi_\pi = \pi)$ belong to the same band, whereas $\gamma = 0$ if $\omega(\phi_0 = 0)$ and $\omega(\phi_\pi = 0)$ are in the same band. Interestingly, $\gamma$ encounters a topological transition induced by the critical amplitude $A = A_c$ if the frequencies merge at $\omega(\phi_\pi = 0, A_c) = \omega(\phi_\pi = \pi, A_c)$. This transition is exemplified by the minimal model of nonlinear topological lattice in Sec.III. It is worth emphasizing that despite all the discussions of nonlinear Schrödinger equations and the quantization of Berry phase, the model is purely classical in the sense of $\hbar$ being zero.

An intuitive way to understand the quantization of nonlinear Berry phase is to compare with Berry phase in linear systems, $\gamma_{linear} = \oint_{BZ} dq (|\Psi_q^{(2)}|^2 \partial_q \phi_q + i(\Psi_q^{(1)}, \Psi_q^{(2)})^* \partial_q (\Psi_q^{(1)}, \Psi_q^{(2)})^\top)$. Under reflection symmetry, the second term in $\gamma_{linear}$ vanishes, and the eigenmode components yield $|\Psi_q^{(1)}| = |\Psi_q^{(2)}|$ to quantize the first term in $\gamma_{linear}$. Likewise, in nonlinear Berry phase of Eq. (3), the second term in the numerator is vanished by reflection symmetry, and the first term picks the quantized integer value due to reflection symmetry constraints in Eq. (5).

Based on the quantized nature of the topological number, it is natural to expect that $\gamma$ is invariant under weak nonlinearity. This is demonstrated in SI. III.A using the perturbation theory called method of multiple-scale[32,33,54,70]. In the strongly nonlinear regime, $\gamma$ still manifests stability against mode disturbances by staying as the integer. Corresponding demonstrations are carried out in SI. II.B.

Having established quantized Berry phase, we now search additional properties for vanishing on-site potential, $\epsilon_0 = 0$. In the linear limit, the model respects chiral symmetry[4,5], which demands that the eigenstates appear in $\pm\omega$ pairs, and the topological mode have zero-energy. To have $\pm\omega$ pairs of nonlinear modes, we require the parity of the interactions to satisfy $f_i(x, y) = f_i(-x, y) = -f_i(x, -y)$. Consequently, the system is invariant under the transformation $(\Psi_n^{(1)}(\omega t), \Psi_n^{(2)}(\omega t)) \rightarrow (-\Psi_n^{(1)}(-\omega t), \Psi_n^{(2)}(-\omega t))$. Given a nonlinear mode $\Psi_\omega$ defined in Eq. (2), this transformation demands a partner solution $\Psi_{-\omega} = (-\Psi_q^{(1)}(-\omega t - qn), \Psi_n^{(2)}(-\omega t - qn + \phi_q))^\top$. Therefore, the frequencies of nonlinear bulk modes appear in $\pm\omega$ pairs. As shown in SI. IV.D, the frequencies of nonlinear topological modes are guaranteed to be zero, which is the nonlinear extension of static topological edge modes in chiral-symmetric systems[4,5].

Topological transition and bulk-boundary correspondence in the minimal model: We now clarify the nonlinear extension of bulk-boundary correspondence[26,71] by demonstrating topological edge modes in the minimal model that respects time-reversal symmetry, where the couplings are specified as

$$f_i(x, y) = c_i y + d_i[(\mathrm{Re} y)^3 + i(\mathrm{Im} y)^3], \quad (7)$$

with $c_i, d_i > 0$ for $i = 1, 2$. This interaction offers numerically stable nonlinear bulk and topological edge modes, and it can be realized in active electrical circuits (Sec. IV and SI. V).

We are interested in attributes unique to nonlinear systems, in particular the topological phase transition induced by bulk mode amplitudes. Thus, the parameters yield $c_1 < c_2$ and $d_1 > d_2$ ($c_1 > c_2$ and $d_1 < d_2$) to induce topological-to-non-topological phase

transition (non-topological-to-topological transition) as amplitudes increase. We abbreviate them as "T-to-N" and "N-to-T" transitions, and they are converted to one another by simply flipping intracell and intercell couplings. In the remainder of this paper, a semi-infinite lattice subjected to open boundary condition (OBC) is always considered whenever we refer to topological edge modes.

We first study the case $c_1 < c_2$ and $d_1 > d_2$, in which a T-to-N transition occurs. Figure 1e numerically illustrates nonlinear band structures and topological transition by considering $\epsilon_0 = 1.5, c_1 = 0.25, c_2 = 0.37, d_1 = 0.22$, and $d_2 = 0.02$. Given that Berry phase $\gamma(A = 0) = \pi$, the lattice is topologically nontrivial in the linear limit. As amplitudes rise, the topological invariant $\gamma(A < A_c) = \pi$ cannot change until it becomes ill-defined when the nonlinear band gap closes at the transition amplitude $A_c$. The band gap reopens above $A_c$, allowing the well-defined Berry phase to take the trivial value $\gamma(A > A_c) = 0$, as depicted in the inset of Fig. 1e. $A_c$ is numerically computed by solving the band gap-closing equation $\omega(\phi_\pi = 0, A_c) = \omega(\phi_\pi = \pi, A_c)$. We propose a convenient approximation[72] $f(\Psi_{n'}^{(j)}, \Psi_n^{(j)}) \approx (c_i + \tfrac{3}{4} d_i A^2) \Psi_n^{(j)}$ to estimate the transition amplitude $A_c \approx \sqrt{-4(c_2 - c_1)/3(d_2 - d_1)}$. The good agreement between this approximation and the numerical solutions is shown in SI. Fig. 6c. We highlight that $A_c^2 \max(d_1, d_2)/\max(c_1, c_2) \approx 0.5$, which demonstrates the comparable nonlinear and linear interactions in the strongly nonlinear regime.

Bulk-boundary correspondence has been extended to weakly nonlinear Newtonian[26] and Schrödinger[71] systems by showing topological boundary modes guaranteed by topologically nontrivial Berry phase. In the strongly nonlinear problem, we utilize analytic approximation and numerical experiment, to doubly confirm this correspondence by identifying nonlinear topological edge modes. In the former, the lattice is composed of $N = 45$ unit cells with OBCs on both ends to mimic semi-infinite lattice, and the parameters are carried over from Fig. 1. The topological mode and frequency are denoted as $\Psi_n = (\Psi_n^{(1)}, \Psi_n^{(2)})^\top$ and $\omega_T$, respectively. Analogous to linear SSH chain[55], the analytic scheme is to approximate $\Psi_n^{(1)} \gg \Psi_n^{(2)}$, which is numerically verified in Fig. 2d. We make one further approximation to truncate the equations of motion to fundamental harmonics. Therefore, the nonlinear topological edge mode is approximated as $\Psi_n \approx (\psi_{1,n}^{(1)}, 0)^\top e^{-i\epsilon_0 t}$, where $\psi_{1,n}^{(1)}$ are the fundamental harmonic components. By doing so, we find $\omega_T = \epsilon_0$, and

$$\left(c_1 + \tfrac{3}{4} d_1 |\psi_{1,n}^{(1)}|^2\right) |\psi_{1,n}^{(1)}| = \left(c_2 + \tfrac{3}{4} d_2 |\psi_{1,n+1}^{(1)}|^2\right) |\psi_{1,n+1}^{(1)}|. \quad (8)$$

From Eq. (8), the semi-infinite lattice hosts topological evanescent modes when $|\Psi_1^{(1)}| < \sqrt{-4(c_2 - c_1)/3(d_2 - d_1)} \approx A_c$, whereas no such mode exists for $|\Psi_1^{(1)}| > \sqrt{-4(c_2 - c_1)/3(d_2 - d_1)} \approx A_c$. The frequency and analytic expression are applied in weakly nonlinear regime (see SI. IV.B), and they are perfectly in line with method of multiple-scale[32,33,54,70]. The numerical scenario is accomplished by applying a Gaussian profile signal $S_n = \delta_{n1} S e^{-i\omega_{ext} t - (t-t_0)^2/\tau^2} (1, 0)^\top$ on the first site, where the carrier frequency $\omega_{ext} = \epsilon_0 = 1.5, T = 2\pi/\omega_{ext}, \tau = 3T$ controls Gaussian spread, and $t_0 = 15T$ denotes trigger time. Figure 2b and f together verify bulk-boundary correspondence[26,71] by identifying the presence and absence of topological boundary excitations below and above the critical amplitude $A_c$, respectively. In Fig. 2(d), the flattened part near the lattice boundary is the manifestation of nonlinearities. These nonlinear topological states are stable against mode disturbances, which is mathematically demonstrated in SI. IV.A.

One may find it unusual that the frequencies of topological modes $\omega_T = \epsilon_0$ are independent of amplitudes, although this result is in agreement with Refs. [7,26,48] in weakly nonlinear regime

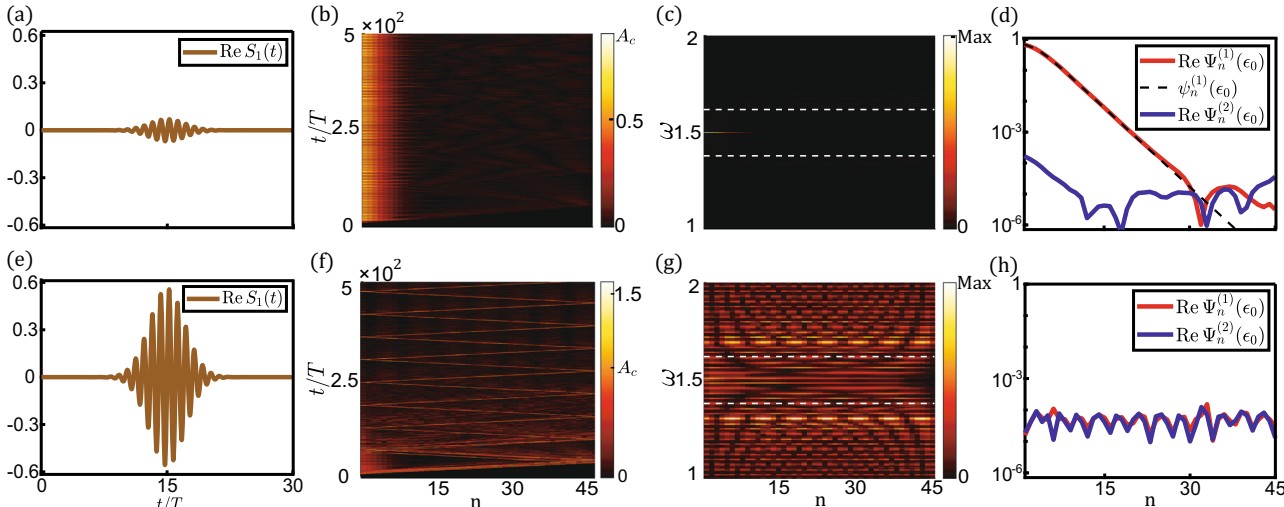

**Fig. 2 Nonlinear edge excitations of the model subjected to T-to-N transition, where the parameters fulfill $c_1 < c_2$ and $d_1 > d_2$. a–d** and **e–h** show lattice boundary responses in small-amplitude topological regime and large-amplitude nontopological regime, respectively. The magnitudes of Gaussian tone bursts are $S = 7 \times 10^{-2}$ in (**a**) and $S = 56 \times 10^{-2}$ in (**e**), respectively. (**b**) and (**f**), spatial-temporal profiles of $|Re\Psi_n^{(1)}(t)|$ for all 45 sites, where $|Re\Psi_n^{(1)}(t)|$ denote the strength of the lattice excitations. (**c**) and (**g**), spatial profiles of the frequency spectra of the responding modes, where the time domain of performing Fourier analysis is from $250T$ to $500T$. White dashed lines mark the top and bottom of the linear band gap. In (**g**), modes in the band gap are triggered by energy absorption[54] from nonlinear bulk modes. (**d**) and (**h**), red and blue curves for the spatial profiles of the $\omega = \epsilon_0$ wave component of the excitations. The analytic prediction of the topological mode $\psi_n^{(1)}(\epsilon_0)$ is depicted by the black dashed curve in (**d**).

and is proved in SI. IV.A. Here we propose an explanation for this intriguing result. Because the evanescent mode fades to zero in the bulk, the "tail" of this mode eventually enters into the small-amplitude regime where nonlinearities are negligible and linear analysis becomes effective. Linear topological theory[55] demands the tail of the mode to be $\omega_T = \epsilon_0$, which in turn requires the frequency of the nonlinear topological mode to be independent of the amplitude.

Topological protection is featured in multiple aspects. As visualized in Fig. 1e, the frequencies of topological modes stay in the band gap and are distinct from nonlinear bulk modes. The appearance and absence of these modes are captured by the topological invariant that cannot change continuously upon the variation of system parameters. Lastly, topological modes are insensitive to defects, which is numerically verified in SI. IV.B.

In the second case of $c_1 > c_2$ and $d_1 < d_2$, N-to-T (non-topo-logical-to-topological) transition occurs as amplitudes rise. We exemplify boundary excitations in Fig. 3 by letting $\epsilon_0 = 8$, $c_1 = 0.37$, $c_2 = 0.25$, $d_1 = 0.02$, and $d_2 = 0.22$. A Gaussian signal is applied on the first site of the lattice, where the carrier frequency $\omega_{ext} = \epsilon_0 = 8$, $T = 2\pi/\omega_{ext}$, Gaussian spread $\tau = 10T$, and trigger time $t_0 = 25T$. In the small-amplitude regime, we consider a chain of $N = 45$ unit cells. As shown in Fig. 3b, the lattice is free of topological modes for $|\Psi_1^{(1)}| < A_c = 0.8944$. In the large-amplitude regime, the lattice is constructed from $N = 120$ unit cells. Anomalous topological edge modes emerge when $|\Psi_1^{(1)}| > A_c$ (see Fig. 3f, h). In contrast to conventional topological modes that shrink to zero over space, $\Psi_n^{(1)}$ decay to the plateau $A_c$ governed by the stable fixed point of Eq. (8), whereas $\Psi_n^{(2)}$ increase to $A_c$ by absorbing energy[54] from $\Psi_n^{(1)}$. Theoretical analysis predicts that the plateau should extend to infinity, but the plateau is limited to reach site 60 by the finite lifetime of topological modes due to the energy conversion to bulk modes, as elaborated in SI. Fig. 5. Despite the huge nonlinearities ($|\Psi_1^{(1)}|/A_c \sim 10$, and $|\Psi_1^{(1)}|^2 max(d_1, d_2)/max(c_1, c_2) \sim 10$), this mode is stable within the finite lifetime of more than 400 periods. The anomalous

behaviors of topological edge states are analogous to those in Refs. [7,26] in which self-induced topological transition is derived in beyond-Kerr weakly nonlinear metamaterials by enabling perturbation theory. Here, the self-induced topological phase is extended to the strongly nonlinear regime and is precisely characterized by the topological number in Eq. (3). This model serves as the combined prototype of long-lifetime, high-energy storage, long-distance transmission of topological modes, and efficient frequency converter from Gaussian inputs to monochromatic signals.

Although T-to-N and N-to-T transitions are converted to one another by choosing the unit cell, topological modes behave qualitatively different (Fig. 2d and 3h) due to the distinction in the fixed points of Eq. (8). The modes converge to the stable fixed point 0 in T-to-N transition ($A_c$ in N-to-T transition), but this fixed point becomes unstable in N-to-T transition (T-to-N transition).

Finally, it is important to emphasize that these strongly nonlinear in-gap states are symmetry-protected topological modes, in the sense that they cannot exist in systems with broken reflection symmetry. Berry phase cannot be derived from Eq. (3) to Eq. (6) and is not quantized for broken reflection symmetry. To demonstrate this, Fig. 4a–d break reflection symmetry by replacing $\epsilon_0$ with $\epsilon_A = (1 + 5\%)\epsilon_0$ and $\epsilon_B = (1-5\%)\epsilon_0$ for A and B-sites, respectively, leading to the disappearance of in-gap nonlinear boundary excitations. Instead of violating reflection symmetry, we introduce disorders by changing the coupling coefficients to $c_{i,n} = c_i + \delta c_{i,n}$ and $d_{i,n} = d_i + \delta d_{i,n}$, where $\delta c_{i,n}/c_i$ and $\delta d_{i,n}/d_i \in [-10\%, +10\%]$ are random variables for different unit cells. As shown in Fig. 4e–h, in-gap nonlinear topological states are robust against disorders by manifesting themselves on the lattice open boundary.

Proposals for experimental implementations: Upon establishing nonlinear topological band theory, it is natural to ask if any realistic physical systems have these properties. Using the symmetry-based analytic methodology in Eq. (4), recent work[73]

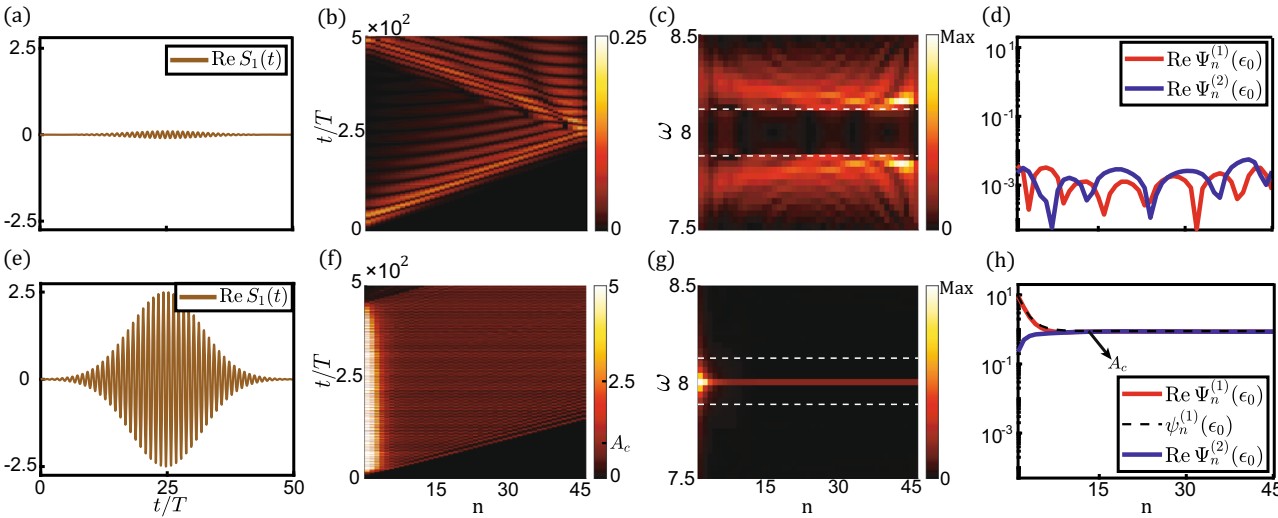

**Fig. 3 Nonlinear boundary responses of the lattice subjected to N-to-T transition, where the parameters yield $c_1 > c_2$ and $d_1 < d_2$. a–d** and **e–h** exhibit lattice boundary excitations in the small-amplitude non-topological regime and the large-amplitude topological regime, respectively. The magnitudes of Gaussian signals are $S = 0.1$ in (**a**) and $S = 2.5$ in (**e**), respectively. (**b**) and (**f**), spatial-temporal profiles of $|\mathrm{Re}\Psi_n^{(1)}(t)|$ for 45 sites. (**c**) and (**g**), frequency spectra of the lattice excitations for 45 sites. Fourier analysis is executed from $250T$ to $500T$. White dashed lines encircle the linear band gap. (**d**) and (**h**), red and blue curves for the spatial distributions of the $\omega = \epsilon_0$ mode component of the lattice excitations. The analytic result of the anomalous topological modes $\psi_n^{(1)}(\epsilon_0)$ is captured by the black dashed curve in (**h**).

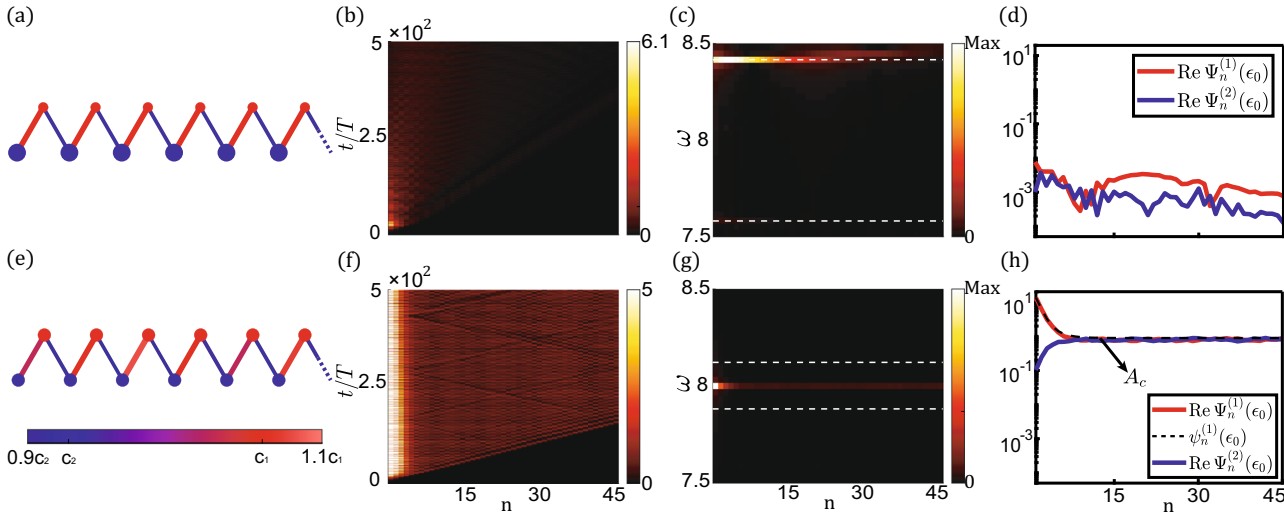

**Fig. 4 Nonlinear boundary responses of the lattices in which reflection symmetry is broken in (a,b,c,d), and disorders are introduced in (e, f, g, h).** $\epsilon_0$, $c_1, d_1, c_2, d_2$ are carried over from Fig. 3, and the external Gaussian signal is adopted from Fig. 3(e). (**a**) and (**e**), the lattices are composed of $N = 120$ unit cells with OBCs on both ends. Reflection symmetry is broken in (**a**) by letting $\epsilon_A = (1 + 5\%)\epsilon_0$ and $\epsilon_B = (1–5\%)\epsilon_0$, which are represented by big blue and small red dots, respectively. Disorders are introduced in (**e**) by random parameters $c_i + \delta c_{i,n}$ and $d_i + \delta d_{i,n}$. Here, the linear random couplings are depicted both by color and line width of the bonds. (**b**) and (**f**), spatial-temporal profiles of $|\mathrm{Re}\Psi_n^{(1)}(t)|$ for 45 sites. (**c**) and (**g**), Fourier analysis is performed from $250T$ to $500T$ to have the spatial-frequency profiles of (**b**) and (**f**). (**d**) and (**h**), the spatial profiles of the $\omega = \epsilon_0$ component are represented by red and blue curves. Black dashed curve in (**h**) is the analytic result of the anomalous topological mode $\psi_n^{(1)}(\epsilon_0)$.

extends the study of topological mechanics to strongly nonlinear regime and manifests strongly nonlinear topological boundary dynamics. Here we discuss an alternative example, namely the active topoelectrical circuit, that manifests topological nonlinear boundary excitations. This experimental proposal is modified from the cascaded circuit ladder by Hadad, et. al., where the nonlinear capacitors in Ref. [7] are now replaced by linear ones, and the nonlinearity in our model is introduced by active voltage sources (Fig. 5). The unit cell is composed of two *LCR* resonators

and two linear capacitors $C_1$ and $C_2$. The inductances are connected to external active voltage sources $\delta V_n^{(1)}$ and $\delta V_n^{(2)}$. These external sources are nonlinearly controlled by $V_n^{(1)}$ and $V_n^{(2)}$, which are the voltage fields of the *LCR* resonators.

Without these nonlinear voltage sources (i.e., let $\delta V_n^{(1)} = \delta V_n^{(2)} = 0$ for all $n$), the linear circuit system manifests topological boundary voltage excitations due to the reflection-symmetric nature of the dynamics. Thus, it is intuitive to expect that topological boundary voltages also arise as long as the nonlinear

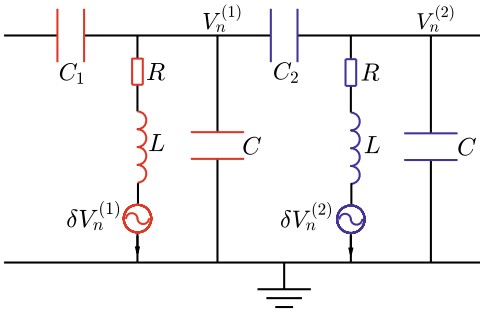

**Fig. 5 Experimental proposal of the nonlinear topoelectrical metamaterial.** The unit cell of the active topological circuit is depicted by the figure. The inductances are connected by the external alternating voltage sources $\delta V_n^{(j=1,2)}$, which are nonlinearly controlled by the voltage fields $V_n^{(j=1,2)}$.

voltage sources respect reflection symmetry. To this end, we demand that the nonlinear sources yield the constraints $\delta V_n^{(1)} = \delta V_n^{(1)}(V_n^{(2)}, V_{n-1}^{(2)})$ and $\delta V_n^{(2)} = \delta V_n^{(1)}(V_n^{(1)}, V_{n+1}^{(1)})$, but their functional forms can be arbitrary. The motion equations of the voltage fields $V_n^{(j=1,2)}$ are captured by Eq. (1), which are elaborated in SI. V. Based on the methodology in Eqs. (1) and (5), the topological protection of the ladder circuit is demonstrated by quantized nonlinear Berry phase.

## Discussion

In this paper, we extend topological band theory to strongly nonlinear Schrödinger equations beyond Kerr-type nonlinearities. The proper definition of Berry phase is carried out for nonlinear bulk modes, and its quantization is demonstrated in reflection-symmetric models. The topological invariant experiences transitions induced by mode amplitudes. These results can be extended to higher-dimensional systems with arbitrarily complex unit cells, but we leave the full proof for the future. Higher-order nonlinear topological hinge states[74] can be followed from these works.

The advent (disappearance) of topological modes is associated with a change in the Berry phase to its topological (non-topological) value. As amplitudes increase, T-to-N (topological-to-non-topological) and N-to-T (non-topological-to-topological) transitions take place for different choices of unit cells. Anomalous topological modes decrease away from lattice boundaries to a plateau controlled by the stable fixed point of nonlinearities. These two unconventional properties stem from the interplay between nonlinear dynamics and topological physics.

By comparing our results with recent developments regarding topological attributes in classical structures[43,46,47,53], we discuss future directions of nonlinear topological physics. These existing literature mainly belong to two classes, namely the topological numbers and properties in Kerr-nonlinear systems[43,46,47], and stochastic topological dynamics in bio-physical cycles[53].

With regards to Kerr-nonlinear topological systems, previous works[46,47] study the nonlinear effects in adiabatic geometric phases. Besides the common interest in nonlinear topological phases, there are several differences between Refs. [46,47] and our work. In particular, Ref. [46] conducts a 2D Kerr-nonlinear Chern insulator, where the geometric Zak phase is naturally unquantized due to the lack of symmetry. Interestingly, Ref. [46] figures the quantized Aharonov-Bohm phase that characterizes nonlinear Dirac cones. It is therefore intriguing to ask if "beyond-Kerr" interactions can also realize nonlinear Dirac cones protected by quantized Aharonov-Bohm phase and enable novel designs of nonlinear mechanical and electrical metamaterials. Ref. [47] studies the interplay between chiral symmetry and

topological attributes in 1D Kerr-nonlinear systems. Following this idea, it is worth asking how chiral symmetry quantizes Berry phase and reveals novel topological physics in general nonlinear systems. Ref. [43] studies the topological phases of Kerr-nonlinear 3D photonic metamaterial, where the topological invariant is the 3-form Chern-Simons theory. Following this idea[43], it is worth exploring 3D topological insulators with general nonlinearities using the formalism of our work.

We now discuss the interplay between nonlinear topological physics and biological dynamics. Ref. [53] studies topological properties in a linear non-equilibrium stochastic process, where complex eigenfrequencies manifest themselves in the non-Hermitian system. Thus, it is exciting to investigate how topologically robust nonlinear edge flow arises in nonlinear active biological cycles, which demands a systematic construction of non-Hermitian, nonlinear topological band theory. Moreover, bulk-boundary correspondence and edge distribution of biomass[37,53,75] may be further identified using the invariant derived in this work.

## Methods

Our primary methods were analytical theories accompanied by computer-aided simulations on MATLAB interface. Nonlinear bulk modes were numerically computed by shooting method. The stability of nonlinear modes were confirmed by the analytic Floquet analysis and the numerical integration algorithm. The algorithmic details are displayed in the supplementary information.

## Data availability

The authors declare that the primary data supporting the findings of this study are enumerated within the article and the supplementary information. The subsequent data generated by the primary data and the MATLAB code have been made freely available on GitHub (https://github.com/kQqr/nonlinearBerryPhase).

## Code availability

The numerical data of the nonlinear modes were generated by shooting method on MATLAB interface. We have made all MATLAB codes freely accessible on GitHub (https://github.com/kQqr/nonlinearBerryPhase).

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

## Acknowledgements

D. Z. would like to thank insightful discussions with Xueda Wen, Junyi Zhang, Changyin Ji, and Feng Li. The work is supported by the the NSF of China (Grants Nos. 11734003, 12061131002), National Key R&D Program of China (Grant No. 2020YFA0308800), and the Strategic Priority Research Program of Chinese Academy of Sciences (Grant No. XDB30000000).

## Author contributions

D.Z. conceived this project. Y.Y. and D.Z. supervised the project. D.Z. wrote the numerical code, and did all calculations. D.Z., D.Z.R., M.L., and Y.Y. constructed the theoretical model, discussed the results, analyzed the data, and contributed to the preparation of the manuscript.

## Competing interests

The authors declare no competing interests.
