## [Peer Review File · Nature Communications]

Topological Invariant and Anomalous Edge Modes of Strongly Nonlinear SystemsREVIEWER COMMENTS

Reviewer #1 (Remarks to the Author):

The authors study a topological number for nonlinear systems. The study on topological physics in nonlinear systems is an interesting new field. In this context, this paper has a potentiality to be published in Nature Communications. However, there are several concerns on the validity of the results. The authors should properly answers the following questions.

1)

The authors should summarize the current understanding of nonlinear topological numbers. For example, the nonlinear topological number is discussed in Thomas Tuloup, Raditya Weda Bomantara, Ching Hua Lee and Jiangbin Gong, arXiv:2006.09753; Raditya Weda Bomantara, Wenlei Zhao, Longwen Zhou, and Jiangbin Gong, Phys. Rev. B 96, 121406(R) (2017). What is the relevance of the present paper to these papers? Is it identical or not? What is the merit and the demerit comparing to it? Are there any other proposals to define topological numbers in nonlinear systems?

2)

Quantization of a topological number is assured by homotopy. For example, the winding number is described by the homotopy $\pi_1(S^1)$. In general, a topological number is related by a homotopy $\pi_d(M)$, where d is the dimension of the system and M is a matrix describing the system. On the other hand, I do not know how to characterize a homotopy in nonlinear systems. The authors should clearly discuss a relevant homotopy for nonlinear systems.

3)

Is there a topological phase transition induced by nonlinearity? Namely, does the nonlinear topological number jump by increasing the nonlinearity? If so, it is possible to derive it analytically? The authors should show numerical results based on Eq.(3), where it is quantized and jumps as a function of a certain parameter.

4)

It is expected that the topological phase is robust against weak nonlinearity. Is it possible to show that the topological number does not change for weak nonlinearity by using a perturbation theory?

5)

How can we obtain Ψ defined by (2). Is it obtained by numerically solving (1)? I expect that the initial condition is crucial to determine Ψ . What is the initial condition?

6)

Is the ansatz (2) stating the periodicity of Ψ always correct? How is the ansatz (2) universal? Is there any solution which is not written in the form of (2)?

7)

Is there any new characteristic feature in the model (7) comparing to the nonlinear SSH model?
Is there any new physics in a generalized nonlinear Schroedinger equation which is absent in the original nonlinear Schroedinger equation?

8)

The authors should present an intuitive picture why the topological number defined in (3) is quantized.

9)

In the SSH model, chiral symmetry is essential. However, chiral symmetry is not discussed in this paper.

The authors should discuss the role of chiral symmetry in strong nonlinear systems.

On the other hand, time-reversal symmetry and reflection symmetry are not important in the SSH model.

Why are these symmetries important for the quantization of the topological number in the present model?

10)

The authors should show the bulk-edge correspondence in strong nonlinear systems.

What is the definition of the topological edge states in strong nonlinear systems?

Is it possible to derive the nonlinear topological edge states analytically?

11)

The authors show a nonlinear band structure for bulk.

Is it possible to show a nonlinear band structure for a finite chain, where the nonlinear edge states are manifest?

12)

Is it possible to characterize a self-induced transition using the topological number presented in this paper, which is a typical phenomenon in nonlinear systems? See Refs.26 and 44.

13)

It is not clear why the electric circuit shown in Fig.5 is nonlinear because all elements are linear.

The derivation of Eq.(5.11) in SM is also not clear.

The authors should add more detailed derivations.

Reviewer #2 (Remarks to the Author):

The manuscript describes topological aspects of a nonlinear Schroedinger equation in a two-component system. The work builds on a recent line of development that has found interest in many areas of physics and engineering, and is very interesting. It appears to have been carried out carefully, and the results are all sound.

I only have a comment regarding the stability of the presented topological features with respect to the wave-like nature of the bulk solution. The justification used by the authors to assume the pure wavelike form for the bulk solution is based on their computational observations in a number of models, presented in the supplement. However, some of those results show significant additional components to the pure wave. What would be interesting to see is a perturbation analysis of the nonlinear equation around the wavelike solution, and an analysis of the stability of the topological features with respect to the new linear contributions. Can this stability analysis also lead to new information about the symmetry requirements? Such an additional analysis will enhance the work of the authors significantly.

My other query will be with regards to the choice of 1D. How robust is the behavior reported by the authors with respect to the choice of dimensionality? For example, can they comment on what would happen if they considered a 2D system, such as the stochastic model studied in Tang et al, PRX 11, 031015 (2021)?

Reviewer #3 (Remarks to the Author):

In this review, I am commenting on the submitted article "Topological Invariant and Anomalous Edge States of Strongly Nonlinear Systems" by Zhou et al. Unfortunately, I will keep my analysis of the issues short in the hopes that the authors may reconsider the form and structure of their presentation. Simply put, I do not find the work interesting, at all. There is not enough context for one who to be able to identify the reason that this work is interesting far less understand how it represents an advance in the field that changes the manner in which I view the field of topology. The work begins in a highly mathematical manner and it continues to be so for the duration of the paper providing little insight into approximations or limitations. Very early in the manuscript, there is a mention that the work will not address the turbulent regime, but it is not really clear what regime is being addressed here/ The work does not, until in a very passing manner, describe anything that resembles a physical system and so it is not possible to make the connection between the physical nature of a system and the mathematics that is presented. As the presented work touches on circuit models, then, presumably, there would have been an easy connection with the significant literature on topological circuits that currently exists. Furthermore, from a purely practical perspective, the interest in this work, as currently written, is very clearly limited to those who are theoretically inclined as I cannot see how an experimentalist is going to wade through this math.

In the end, maybe there is something interesting here but I certainly cannot see it. The manuscript does not change the manner in which I view the field of topology broadly nor does it give me additional insights into a system that has been the canonical workhorse for new models. I cannot recommend publishing this work.

Dear Editor and Referees,

We appreciate your careful consideration of our manuscript, “Topological Invariant and Anomalous Edge States of Strongly Nonlinear Systems”. Upon reviewing the referee reports we conclude that the deficiencies identified are addressable and the suggestions are constructive. Therefore, we strive to incorporate the suggestions by the referees and hereby resubmit our revised manuscript. Here are our point-by-point responses to the specific questions and comments from the referees:

I. REPLY TO REFEREE 1

“The authors study a topological number for nonlinear systems. The study on topological physics in nonlinear systems is an interesting new field. In this context, this paper has a potentiality to be published in Nature Communications. However, there are several concerns on the validity of the results. The authors should properly answers the following questions.”

We appreciate the positive recommendation of the referee. Below we respond to the specific suggestions from the referee. We use “Eq.R(X) and Fig.R(X)” to refer to the equations and figures in this reply, and use “Eq.(X) and Fig.(X)” to refer to the equations and figures that appear in the citing references. The referee’s questions and comments are quoted in **blue**, with our responses given in **black** and any text quoting or referencing relevant changes we’ve made given in **bold**. At the end of each answer, we summarize the modifications in the revised manuscript.

“1) The authors should summarize the current understanding of nonlinear topological numbers. For example, the nonlinear topological number is discussed in Thomas Tuloup, Raditya Weda Bomantara, Ching Hua Lee and Jiangbin Gong, arXiv:2006.09753; Raditya Weda Bomantara, Wenlei Zhao, Longwen Zhou, and Jiangbin Gong, Phys. Rev. B 96, 121406(R) (2017).

(1.1) What is the relevance of the present paper to these papers?

(1.2) Is it identical or not?

(1.3) What is the merit and the demerit comparing to it?

(1.4) Are there any other proposals to define topological numbers in nonlinear systems?”

We appreciate the referee for pointing out these two important papers[1,2] that are very much related to our research[3]. The first paper “nonlinear Dirac cones”[1] by Bomantara et. al. (Phys. Rev. B **96**, 121406(R) (2017)) studies un-quantized Berry phase and quantized AB-phase in a Chern insulator with mean field on-site nonlinearities. The second paper “nonlinearity induced topological physics in momentum space and real space”[2] by Tuloup et. al. (Phys. Rev. B **102**, 115411 (2020)) studies the nonlinear Zak phase in SSH model with chiral-symmetry-breaking terms and on-site Kerr nonlinearities. Below we address point-by-point answers to the specific questions (1.1)~(1.4).

Answer to Question #(1.1): Both of Refs.[1,2] and our work[3] study nonlinear adiabatic geometric phases, topological phase transitions, nonlinear band structures, and localized boundary modes in nonlinear Schrödinger equations. Besides the common interest in nonlinear topological physics, there are several differences between Refs.[1,2] and our work[3].

The differences between Ref.[1] and our manuscript is summarized as follows. First, the dimensionality of the $2D$ Chern insulator in Ref.[1] is different from our 1D system. Second, time-reversal symmetry is broken[4] in Chern insulator of Ref.[1], whereas our model respects time-reversal symmetry. Third, Ref.[1] highlights that nonlinear Dirac cone is protected by nonlinear interactions and are robust against symmetry-violating perturbations. This is different from the conventional linear Dirac cones and topological boundary modes[3] that are protected by symmetries. Fourth, the geometric Berry phase is naturally un-quantized in Ref.[1] due to the lack of symmetry, but Ref.[1] obtains an interesting *quantized* Aharonov-Bohm phase that characterizes nonlinear Dirac cones. Finally, Ref.[1] considers Kerr-nonlinearity for on-site potential, whereas our work studies general nonlinearities for hopping terms that lead to topological phase transitions.

We then summarize the similarities and differences between Ref.[2] and our manuscript[3]. First, the dimensionality is the same for 1D SSH models in [2] and [3], but the Kerr on-site nonlinearity in [2] and general hopping nonlinearities in [3] are different. Ref.[2] studies the nonlinear matrix Hamiltonian, whereas the nonlinear dynamics in our work cannot be written in the matrix representation. Second, Ref.[2] conducts the interplay between chiral symmetry and

nonlinear interactions, whereas our work conducts reflection symmetry. Third, the numerical techniques that solve nonlinear modes in Sec.II.B of Ref.[2] are conceptually similar to shooting method in our work.

Answer to Question #(1.2): Both of the nonlinear Zak phase[1,2] and nonlinear Berry phase[3] study the geometric phase of nonlinear waves acquired by evolving system parameters. They characterize topological invariants and boundary modes for symmetry constraints and nonlinear effects, yet they have differences.

First, both of Refs.[1,2] and our work[3] conduct nonlinear contributions to the geometric phases. Nonlinear Zak phase[1,2] is modified by a kernel \mathcal{K} that stems from the state-dependent Hamiltonian matrix. Our work[3] also conducts the state-dependent dynamics, but the motion equations cannot be expressed by a matrix Hamiltonian. Nonlinear correction of Berry phase is incorporated in the change of wave form as the wave number varies, as shown in the term $\psi_{l,q}^{(j)*} \partial_q \psi_{l,q}^{(j)}$ in Eq.(3) of the main text.

Second, we discuss the difference between the kernel[2] \mathcal{K} and the “change of wave form” $\psi_{l,q}^* \partial_q \psi_{l,q}$. The difference can be understood within a linear SSH model with broken chiral and reflection symmetries. The kernel[2] reduces to $\mathcal{K} = 1$ due to the vanishing nonlinearity, but the “change of wave form” $\Psi_q^* \partial_q \Psi_q$ does not vanish because the eigenstate $|\Psi_q\rangle$ does not stay on the equator of the Bloch sphere. Thus, in nonlinear systems, the term $\psi_{l,q}^* \partial_q \psi_{l,q}$ provides both nonlinear and linear contributions to the adiabatic geometric phase.

Answer to Question #(1.3): There are merits and demerits in our work comparing to Refs.[1,2].

The merits mainly lie in the general nonlinear interactions. First, our work may serve to design general nonlinear topological metamaterials. Second, we show that symmetries can be reflected in nonlinear modes even if their analytic wave forms are unknown. This idea can be extended to other symmetries, such as chiral symmetry discussed in the revised part of Sec.II of the main text[3]. It is intriguing to consider how symmetries harness nonlinear dynamics, such as the stability constraints addressed in the **answers to Question #7** and **Question #1 of Referee 2**. Finally, general nonlinearities may lead to new physics absent in Kerr nonlinearities, as we will discuss in **Question #7**.

Our work has several demerits comparing to Refs.[1,2]. First, our work[3] is limited to 1D systems, whereas Ref.[1] studies 2D nonlinear systems. In **Question #2 of Referee 2**, we introduce a working project[5] that extends current results to 2D systems that host nonlinear Weyl modes. Second, we conduct reflection symmetry rather than chiral symmetry, but chiral symmetry is more essential for linear topological systems. As shown in the **answer to Question #9**, we encounter difficulties using chiral symmetry to quantize Berry phase of general nonlinear systems. Finally, Ref.[2] emerges a “looped” band structure for additional bands as amplitudes grow large. This is not observed in our model because the amplitude range is limited by the stability of nonlinear modes.

Answer to Question #(1.4): There are several references that define topological numbers in nonlinear systems. For example, Ref.[7] defines the topological number for an SSH chain with Kerr-nonlinear hopping strengths. Ref.[8] presents a perturbative analysis on the weakly nonlinear mechanical chain and reduces the “beyond-Kerr” interactions to Kerr-nonlinear couplings. Winding number is defined by Kerr-interactions to characterize weakly nonlinear topological phases and mechanical boundary modes. Ref.[9] proposes on-site Kerr-nonlinear Haldane model that manifests integer-valued Chern number and amplitude-controlled chiral edge modes.

Summary of the modifications: We thank the referee for raising these two important references. In the revised main text, we add a new section titled “summary and perspectives” in Sec.V, Page. 8, marked by blue: Together with the **answer to Question #2 of Referee 2**, we discuss prospective interesting directions of nonlinear topological physics inspired by Refs.[1,2,26].

“2) Quantization of a topological number is assured by homotopy. For example, the winding number is described by the homotopy $\pi_1(S^1)$. In general, a topological number is related by a homotopy $\pi_d(M)$, where d is the dimension of the system and M is a matrix describing the system. On the other hand, I do not know how to characterize a homotopy in nonlinear systems. The authors should clearly discuss a relevant homotopy for nonlinear systems.”

We appreciate this insightful question raised by the referee. In linear chiral-symmetric SSH model with the Hamiltonian $H = \vec{h} \cdot \vec{\sigma}$, quantization of Berry phase is assured by the homotopy $\pi_1(S^1)$ that maps from the 1D Brillouin zone to the “magnetic field” \vec{h} constrained on the equator of the Bloch sphere. Likewise, quantization of nonlinear Berry phase is guaranteed by mapping the Brillouin zone $q \in [-\pi, \pi]$ to the “magnetic field $\vec{h}(\Psi)$ ” constructed from nonlinear bulk modes Ψ . $\vec{h}(\Psi)$ is constrained by reflection symmetry on the $(h_x, h_y, h_z = 0)$ 2D horizontal plane. Thus, the mapping from 1D Brillouin zone to the closed trajectory $\vec{h}(\Psi)$ in 2D horizontal plane picks quantized integer values.

The aforementioned magnetic field $\vec{h}(\Psi)$ is constructed in the following way. By recalling nonlinear bulk modes

$\Psi_q = (\Psi_q^{(1)}(\theta), \Psi_q^{(2)}(\theta + \phi_q))$ in Eq.(2) of the main text, we define the following two functions

$$\begin{aligned} Z_q^{(1)} &= [\Psi_{+q}^{(1)}(-\phi_{+q}) + \Psi_{-q}^{(1)*}(-\phi_{-q})]/2, \\ Z_q^{(2)} &= [\Psi_{-q}^{(2)}(+\phi_{-q}) + \Psi_{+q}^{(2)*}(+\phi_{+q})]/2. \end{aligned} \quad (1)$$

In purely linear SSH model, the wave components are given by $\Psi_q^{(1)}(\theta) = e^{-i\theta} \cos(\theta_q/2)$ and $\Psi_q^{(2)}(\theta) = e^{-i\theta} \sin(\theta_q/2)$, and the functions in Eqs.R(1) are simplified as $Z_q^{(1)} = e^{i\phi_q} \cos(\theta_q/2)$ and $Z_q^{(2)} = e^{i\phi_q} \sin(\theta_q/2)$. Thus, it becomes clear that $\arg(Z_q^{(1)})$ and $\arg(Z_q^{(2)})$ can characterize the relative phase ϕ_q between the two linear (nonlinear) wave components, and the ratio $Z_q^{(1)}/Z_q^{(2)}$ captures the relative magnitude of the two linear (nonlinear) wave components. Next, we construct the three-component magnetic field $\vec{h}(\Psi)$,

$$\begin{aligned} h_x(\Psi) &= \text{Re} \left[\frac{2Z_q^{(1)} Z_q^{(2)}}{\sqrt{(Z_q^{(1)2} + Z_q^{(2)2})(|Z_q^{(1)}|^2 + |Z_q^{(2)}|^2)}} \right], \\ h_y(\Psi) &= \text{Im} \left[\frac{2Z_q^{(1)} Z_q^{(2)}}{\sqrt{(Z_q^{(1)2} + Z_q^{(2)2})(|Z_q^{(1)}|^2 + |Z_q^{(2)}|^2)}} \right], \\ h_z(\Psi) &= \text{Re} \left[\frac{Z_q^{(1)2} - Z_q^{(2)2}}{Z_q^{(1)2} + Z_q^{(2)2}} \right]. \end{aligned} \quad (2)$$

As the wave number traverses the Brillouin zone, h_x and h_y together depict the winding of $\vec{h}(\Psi)$ trajectory around the point $(0, 0, h_z)$, whereas h_z itself describes the magnitude difference between the two nonlinear mode components. In linear SSH model, we obtain $\vec{h} = (\cos \phi_q \sin \theta_q, \sin \phi_q \sin \theta_q, \cos \theta_q)$ from the eigenstates. Without symmetry constraints, \vec{h} is freely movable on a closed trajectory *not* on the equator of the Bloch sphere. Chiral or reflection symmetry demands that $\theta_q = \pi/2$ and the subsequent $\vec{h} = (\cos \phi_q, \sin \phi_q, 0)$ trajectory is constrained on the equator. Likewise, for nonlinear systems without symmetry properties, $\vec{h}(\Psi)$ can be any trajectory not restricted in the 2D horizontal plane. Reflection symmetry in the nonlinear model demands the wave components to yield $\Psi_q^{(2)} = \Psi_{-q}^{(1)}$ and $\phi_{-q} = -\phi_q$, which in turn offer us the result $Z_q^{(1)} = Z_q^{(2)}$ in Eqs.R(1). Therefore,

$$\vec{h}(\Psi) = (\cos(\arg Z_q^{(1)}), \sin(\arg Z_q^{(1)}), 0) \quad (3)$$

is in consequence a closed 1D circular trajectory embedded in the 2D horizontal plane. The map from 1D Brillouin zone to $\vec{h}(\Psi)$ is depicted by the homotopy group $\pi_1(S^1) = \mathcal{Z}$ that counts how many times $\vec{h}(\Psi)$ winds around the origin. The topological phase of the nonlinear lattice is described by the winding number \mathcal{N} of $\vec{h}(\Psi)$ -trajectory, which is further captured by

$$\mathcal{N} = (\arg Z_\pi^{(1)} - \arg Z_0^{(1)})/\pi = (\phi_\pi - \phi_0)/\pi, \quad (4)$$

where ϕ_π and ϕ_0 are the relative phases of nonlinear mode components at high-symmetry points. The nonlinear system is topologically non-trivial (trivial) if $\phi_\pi - \phi_0 = \pi$ ($\phi_\pi - \phi_0 = 0$). Interestingly, the system experiences a topological phase transition induced by mode amplitudes as $\phi_\pi - \phi_0$ abruptly changes between 0 and π . We will elaborate on this point in the **answer to Question #3** below.

Concluding remarks: As the referee notes, within the context of topological insulators, homotopies are associated with the linear matrix representations of the Hermitian operators that govern their dynamics, which rightly raises questions as to the application of such principles to nonlinear modes. For example, a homotopy might be associated in the linear case with how the phase of a vector winds as one traverses a 1D Brillouin Zone. How, then, to obtain a homotopy for a system that lacks a linear vector space? To do so, it is important to recall that in the more general mathematical formalism, there is nothing specific to linearity in defining homotopies. Rather, homotopies are the in-equivalence classes (for example, those indexed by an integer) associated with ways in which one topological space, often a hyper-sphere, can be mapped onto another. These spaces are not in general linear vector spaces. Homotopies may thus be established in spaces of nonlinear modes provided that we have a mapping analogous to the linear case, in which a point in the Brillouin Zone is mapped onto a complex vector (constructed from the wave function).

Summary of the modifications: We thank the referee for bringing this important question. The above contents are incorporated in the revised supplementary information (abbreviated as ‘‘SI’’ in the rest of this reply) Sec.II.D,

Page.7, marked by blue. The idea is to build the homotopy mapping from Brillouin zone to the “magnetic field” constructed from nonlinear bulk modes.

“3) Is there a topological phase transition induced by nonlinearity? Namely, does the nonlinear topological number jump by increasing the nonlinearity? If so, it is possible to derive it analytically? The authors should show numerical results based on Eq.(3), where it is quantized and jumps as a function of a certain parameter.”

We thank the referee for raising these questions. Indeed, there is a topological phase transition induced by increasing the nonlinearities (mode amplitudes). The quantized integer value of nonlinear Berry phase jumps from π to 0 as the amplitudes A pass through the phase transition point A_c . This transition amplitude can be determined both analytically and numerically by considering the jump in the integer-valued nonlinear Berry phase γ .

First, we analytically show the topological transition as amplitudes rise. We consider the simple case in which the nonlinear bands are always gapped unless they touch at the topological phase transition point. Thus, we focus on the upper nonlinear band to illustrate nonlinear Berry phase. Using reflection symmetry, nonlinear Berry phase is reduced as $\gamma = \phi_\pi - \phi_0$, where ϕ_π and ϕ_0 are the relative phases ϕ_q at high-symmetry points $q = 0, \pi$ of the upper band. They yield quantized values ϕ_π and $\phi_0 = 0$ or $\pi \bmod 2\pi$ due to reflection symmetry constraints, and γ is in consequence quantized. In particular, $\gamma = \pi$ ($\gamma = 0$) if ϕ_π and ϕ_0 pick different (the same) integer values. To determine ϕ_π and ϕ_0 for the upper band, we calculate and compare the nonlinear mode frequencies at high-symmetry points between the upper and lower bands.

We denote $\omega(\phi_{q=0}, A)$ ($\omega(\phi_{q=\pi}, A)$) as the frequency of the nonlinear mode, where the wave number, relative phase, and mode amplitude are $q = 0$ ($q = \pi$), $\phi_{q=0}$ ($\phi_{q=\pi}$), and A , respectively. Given the parameters of our manuscript, $\omega(\phi_{q=0} = 0, A)$ always belongs to the upper band. However, the mode frequency $\omega(\phi_{q=\pi} = \pi, A)$ switches bands for different amplitudes. As shown by the black curve in the inset of the main text Fig.1(e), $\omega(\phi_{q=\pi} = \pi, A)$ belongs to the upper band for $A < A_c$ and γ picks the non-trivial integer value π , whereas $\omega(\phi_{q=\pi} = \pi, A)$ belongs to the lower band for $A > A_c$ and γ picks the trivial value 0.

γ experiences the topological transition induced by the abrupt jump from $\phi_\pi = \pi$ to $\phi_\pi = 0$ as the mode amplitudes A grow beyond the critical value A_c . Meanwhile, the frequency $\omega(\phi_\pi = 0, A > A_c)$ switches to the upper band, $\omega(\phi_\pi = \pi, A > A_c)$ switches to the lower band, and the band inversion occurs. As indicated by the inset of the main text Fig.1(e), A_c is determined when the mode frequencies $\omega(\phi_\pi = \pi, A_c)$ and $\omega(\phi_\pi = 0, A_c)$ meet (the black and green curves meet). Using the analytic forms of high-symmetry-point frequencies derived in SI.III.C, we numerically compute this meeting point between the black and green curves and obtain the topological transition amplitude A_c .

FIG. 1. Numerical computations of γ , $\gamma^{(1)}$, $\gamma^{(2)}$, and $\gamma^{(3)}$ for mode amplitudes ranging from $A = 0.003$ to 1.8. (a) Black and blue lines stand for the amplitude dependences of $\gamma^{(1)}$ and $|\gamma^{(2)}| + |\gamma^{(3)}|$, respectively. (b) Enlarged figure of $|\gamma^{(2)}| + |\gamma^{(3)}|$. (c) Quantized nonlinear Berry phase γ .

We now show numerical results based on Eq.(3) of the main text, where γ is quantized and jumps as amplitudes increase. The numerical strategy is given by SI.II.C. In SI.Eqs.(2.20), nonlinear Berry phase is separated to three parts $\gamma = \gamma^{(1)} + \gamma^{(2)} + \gamma^{(3)}$, and each part is discretized by SI.Eqs.(2.21) for numerical calculations. As shown by the numerical results in Fig.R1(a), a discontinuous jump occurs to $\gamma^{(1)}$ at $A = A_c$, whereas $|\gamma^{(2)}| + |\gamma^{(3)}|$ is negligible. Fig.R1(b) is the enlarged picture of $|\gamma^{(2)}| + |\gamma^{(3)}|$ to state that both $\gamma^{(2)}$ and $\gamma^{(3)}$ randomly fluctuate around 0 due to the numerical tolerance of generating nonlinear bulk states. In Fig.R1(c), nonlinear Berry phase is quantized and discontinuously jumps at the critical amplitude $A = A_c$.

Summary of the modifications: We thank the referee for raising this important point. We discussed these matters in the main text Sec.II, Page.3 right column, and we regret that we were unclear. We have added the following text marked by blue to address this: “Interestingly, γ encounters a topological transition induced by the critical amplitude $A = A_c$ if frequencies merge at $\omega(\phi_\pi = 0, A_c) = \omega(\phi_\pi = \pi, A_c)$. This transition is exemplified by the minimal model of nonlinear topological lattice in Sec.III”. The analytic derivation is given in SI.II.C, Pages.5 and 6, marked by blue. Our approach is to use $\gamma = \phi_\pi - \phi_0$, and compute the high-symmetry relative phases ϕ_π and ϕ_0 by comparing the corresponding frequencies between the upper and lower bands.

We thank the referee for suggesting numerical checks. We perform such checks via the combined methods of shooting method and fast Fourier transformation. The resulting data confirms the analytic prediction of an abrupt topological transition induced by the nonlinearity at a finite mode amplitude. Fig.R1(c) is now shown in the main text, Page.3, right column, Fig.1(d), with the updated figure caption marked by blue. The corresponding numerical details are elaborated in SI.II.C, Page.6, Eqs.(2.20), (2.21). Figs.R1 now appear as Fig.2 in SI.II.C, Page.7, right column.

“4) It is expected that the topological phase is robust against weak nonlinearity. Is it possible to show that the topological number does not change for weak nonlinearity by using a perturbation theory?”

We appreciate the insightful question raised by the referee, and agree that the topological phases are robust against weak nonlinearities. Below, we use the perturbation theory called *method of multiple-scale*[13-16] to demonstrate that (1) the topological number stays as an integer and does not change for weak nonlinearities, and (2) the topological boundary modes are robust against weak nonlinearities.

First, we discuss the robustness of the topological index against weak nonlinearities. Thus, method of multiple-scale is employed to study weakly nonlinear bulk modes. As shown in SI.Eq.(3.6), $\phi_q^{(0)}$ denotes the zeroth-order relative phase between the two components of bulk modes. To the first-order correction of weak nonlinearity (SI.Eq.(3.11)), $\phi_q^{(0)}$ is modified to $\phi_q^{(0)} + \epsilon\phi_q^{(1)}$, where $\phi_0^{(1)} = \phi_\pi^{(1)} = 0$. In SI.Eqs.(3.12) and (3.13), we show that the first-order corrected Berry phase γ is the same as the zeroth-order Berry phase $\gamma = \phi_\pi^{(0)} - \phi_0^{(0)}$. Thus, we demonstrate the invariance of the topological number against weakly nonlinearity.

Second, we use method of multiple-scale to show that topological edge modes are robust against weak nonlinearity. In SI.Eqs.(4.4) and (4.6), both of the zeroth-order and the first-order correction of the edge mode exponentially decay in space, which means that the edge mode localization is insensitive to weak nonlinearity.

Summary of the modifications: The new contents are now updated in the main text, Sec.II, Page.4, left column, Paragraph.3, marked by blue: “Based on the quantized feature of the topological number, it is intuitive to expect that γ is invariant under weak nonlinearity. This is demonstrated in SI.III.A using the perturbation theory called method of multiple-scale.” The analytic derivation is given in SI.III.A, Page.12, marked by blue. Our approach is to compute the Berry phase using the corrected nonlinear bulk modes.

We also perform the robustness check for weakly nonlinear topological edge modes in SI.IV.A, Page.17 marked by blue, where we compute the edge mode corrections in the perturbative manner.

“5) How can we obtain Ψ defined by (2). Is it obtained by numerically solving (1)? I expect that the initial condition is crucial to determine Ψ . What is the initial condition?”

We agree with the referee that $\Psi(t)$ is obtained by numerically solving Eq.(1) of the main text, and the initial condition is crucial to determine $\Psi(t)$.

The numerical *shooting method*[10-12] solves bulk modes in the strongly nonlinear regime. This method computes the difference between the initial and final states of a *trial* wave function after one period. The vanishing difference indicates that the trial wave is a periodic solution. If the difference is non-zero, shooting method evolves the trial wave towards the nonlinear periodic solution by lowering the difference.

The initial condition is determined by weakly nonlinear bulk modes using method of multiple-scale in SI.III.A. Shooting method evolves it into a periodic solution with the small amplitude $A_1 \ll A_c$. Next, we uniformly re-scale the nonlinear wave by a factor slightly greater than 1, and use shooting method to evolve it into a new solution of the amplitude $A_2 > A_1$. By repeating this strategy, we get nonlinear bulk modes for a wide range of amplitudes in the strongly nonlinear regime.

Summary of the modifications: We thank the referee for raising these important questions. In SI.III.B Page.13, left column, Paragraph.3, we add new contents marked by blue for the initial condition of Ψ using method of multiple-scale.

The numerical shooting method is now discussed in the main text, Sec.II, Page.2, right column, Paragraph.2, last sentence, marked by blue: “Eq.(2) is solved by the numerical shooting method that applies for general nonlinearities, as detailed in SI.III.B.”

“6) Is the ansatz (2) stating the periodicity of Ψ always correct? How is the ansatz (2) universal? Is there any solution which is not written in the form of (2)?”

We appreciate the referee for bringing these deep questions. The ansatz in Eq.(2) of the main text always correctly states the periodicity of *nonlinear bulk modes*. However, this ansatz is *not universal* to state the periodicity of other nonlinear modes, such as soliton and nonlinear localized modes.

Three aspects justify the ansatz Eq.(2) for the periodicity of nonlinear bulk modes. First, the ansatz has been demonstrated by method of multiple-scale[13-16] for weakly nonlinear bulk modes. Second, at high-symmetry points, the analytic solutions of strongly nonlinear bulk modes in SI.III.C indicate the correctness of this ansatz. Finally, the numerical shooting method[10-12] manifests non-dispersive, plane-wave like bulk modes in the strongly nonlinear regime, which are perfectly in line with the ansatz in Eq.(2).

However, the ansatz in Eq.(2) is not correct to depict the periodicity of soliton and nonlinear localized modes. Dark and bright soliton modes manifest minimum and maximum amplitudes at the centers of the excitations, respectively, whereas the spatially uniform amplitudes in Eq.(2) violate this feature. Moreover, solitons cannot exist[11,13] under periodic boundary conditions (PBCs), but the ansatz holds true for PBCs. Nonlinear localized modes stem from the bifurcation of nonlinearity. The amplitudes are localized at certain positions of the lattice and the ansatz in Eq.(2) cannot cover these modes.

Summary of the modifications: We thank the referee for raising this important question. The limitation of the ansatz in Eq.(2) is now addressed in the main text. In Sec.II, right column, Paragraph.3 marked by blue: “The ansatz in Eq.(2) correctly states the periodicity of nonlinear bulk modes based on the following reasons.” In Sec.II, right column, Paragraph.4 marked by blue: “While the ansatz in Eq.(2) captures the periodicity of nonlinear bulk states, it cannot describe temporal-periodic nonlinear modes with spatially inhomogeneous amplitudes, such as soliton excitations and nonlinear localized modes. Corresponding discussions are addressed in SI.I.” Finally, we incorporate a more detailed discussion stating the limitation of the ansatz in SI.I, Page.1, marked by blue.

“7) Is there any new characteristic feature in the model (7) comparing to the nonlinear SSH model? Is there any new physics in a generalized nonlinear Schrödinger equation which is absent in the original nonlinear Schrödinger equation?”

We thank the referee for raising these interesting questions. Inspired by the referee’s first question here and **Question #1 of Referee 2**, we manage to find a new characteristic feature in the model (7) comparing to the nonlinear SSH model. We also agree with the referee that there is new physics in generalized nonlinear Schrödinger equations which is absent in the original nonlinear Schrödinger equation, as we discuss below.

First, the new feature in the model (7) is that all nonlinear bulk modes are *marginally stable* within Floquet analysis. As shown in SI.II.E, Eqs.(2.28) and (2.30), the intrinsic symmetries of the model (7) guarantee that the monodromy matrix is real and anti-symmetric in the Floquet stability analysis. All Floquet exponents are *purely imaginary*, and all nonlinear bulk modes are *marginally stable*. This remarkable feature is in stark contrast to the instability in nonlinear modes if the system does not have symmetry constraints.

Second, we discuss new physics in general nonlinear interactions beyond the model (7). Eq.(7) of the main text sets the equal nonlinear coefficients $d_i(\text{Real}) = d_i(\text{Imaginary})$ for real and imaginary field variables. If they are not equal, then real and imaginary field variables have different topological transition amplitudes, such as $A_c(\text{Real}) < A_c(\text{Imaginary})$. It is intriguing to ask which is the true topological transition amplitude, and what happens for amplitudes above $A_c(\text{Real})$ and below $A_c(\text{Imaginary})$. Corresponding new topological physics are yet to be explored in these interesting systems.

Summary of the modifications: We have incorporated the discussions of nonlinear stability in the revised main text Sec.II, Page.3, left column, Paragraph.1 marked by blue: “Due to the symmetry constraints of the nonlinear motion equations, we demonstrate that all nonlinear bulk states are marginally stable within Floquet analysis (see SI.II.E for details).” Detailed discussions of the nonlinear stability are addressed in SI.II.E, Page.8, right column marked by blue, where we conduct the interplay between Floquet analysis and system symmetries to demonstrate marginal stability of nonlinear modes.

“8) The authors should present an intuitive picture why the topological number defined in (3) is quantized.”

We appreciate the constructive suggestion of the referee that greatly improves the accessibility of our work. Two different ways are presented in the revised paper to intuitively understand the quantization of the topological number in Eq.(3).

The first way is to compare with Berry phase in linear SSH model,

$$\gamma_{\text{linear}} = \oint_{\text{BZ}} dq \left[|\Psi_q^{(2)}|^2 \partial_q \phi_q + i(\Psi_q^{(1)}, \Psi_q^{(2)})^* \partial_q \begin{pmatrix} \Psi_q^{(1)} \\ \Psi_q^{(2)} \end{pmatrix} \right], \quad (5)$$

where $|\Psi_q\rangle = (\Psi_q^{(1)}, \Psi_q^{(2)} e^{-i\phi_q})^\top$ is the eigenmode. Under reflection symmetry, the second term in γ_{linear} vanishes, and the eigenmode yields $|\Psi_q^{(1)}| = |\Psi_q^{(2)}|$ to quantize the first term in γ_{linear} . Likewise, in nonlinear Berry phase (Eq.(3) of the main text), the second term in the numerator is vanished by reflection symmetry, and the first term picks the quantized integer value due to the symmetry constraints in Eqs.(5) of the main text.

The second way is to invoke the homotopy group $\pi_1(S^1)$ in the **answer to Question #2**, where we construct the ‘‘magnetic field’’ $\vec{h}(\Psi)$ from nonlinear bulk modes. Without symmetry constraints, the continuous mapping from 1D Brillouin zone to $\vec{h}(\Psi)$ -trajectory can be continuously deformed into a one-point mapping, and so its homotopy class is trivial. With reflection symmetry constraints, $\vec{h}(\Psi)$ -trajectory is restricted on the 2D horizontal plane, and the homotopy mapping from 1D Brillouin zone guarantees the quantization of the nonlinear Berry phase in Eq.(3).

Summary of the modifications: In the main text, Sec.II, Page.4, Paragraph.2 marked by blue, we provide the intuitive way to understand quantized nonlinear Berry phase by relating to the quantization of Berry phase in linear systems. We also incorporate the quantization of nonlinear Berry phase using the homotopy group in SI.II.D, Page.8, left column, Paragraph.2, marked by blue.

‘‘9) In the SSH model, chiral symmetry is essential. However, chiral symmetry is not discussed in this paper. The authors should discuss the role of chiral symmetry in strong nonlinear systems. On the other hand, time-reversal symmetry and reflection symmetry are not important in the SSH model. Why are these symmetries important for the quantization of the topological number in the present model?’’

We agree with the referee’s comment that chiral symmetry is more essential for the topological properties of the SSH model. Below we discuss reflection, chiral, and time-reversal symmetries in details.

We compare the time-dependences of linear and nonlinear bulk modes to state the importance of reflection symmetry for quantizing nonlinear Berry phase. Linear Schrödinger equation is solved by the ansatz $|\Psi_n(t)\rangle = e^{-i\omega_n t} |\Psi_n\rangle$, where ω_n and $|\Psi_n\rangle$ are the n -th eigenvalue and eigenvector, respectively. The ansatz indicates that all eigenmodes are sinusoidal in time and contain fundamental harmonics only. Thus, the ratio $\Psi_n^{(i)}(t)/\Psi_n^{(j)}(t)$ between any two mode components stays constant as t varies. However, bulk modes in general nonlinear systems are composed of all higher-order harmonics. Without symmetry constraints, the time-dependence can be different among mode components, and the ratio $\Psi^{(i)}(t)/\Psi^{(j)}(t)$ is not constant as t varies. In Eq.(3) of the main text, γ contains all Fourier series of the mode components. Without symmetry constraints, these Fourier coefficients are un-correlated and γ is naturally un-quantized. Reflection symmetry constructs the relationships among the Fourier series of mode components to quantize nonlinear Berry phase.

Chiral symmetry is more essential for the topological properties in the linear SSH model. It demands vanishing on-site potential. The eigenfrequencies are constrained to appear in $\pm\omega$ pairs, and the topological modes have zero energy. Likewise, we study chiral symmetry by asking vanishing on-site potential in the nonlinear SSH model. Interestingly, the frequencies of nonlinear bulk modes appear in $\pm\omega$ pairs, and the frequencies of nonlinear topological edge modes are guaranteed to be zero. These fascinating features are the nonlinear extensions of the properties in linear chiral-symmetric systems. However, without reflection symmetry, we encounter difficulties to demonstrate the quantization of γ using chiral symmetry only, because we have not obtained useful relationships among the Fourier series of nonlinear modes. It is our future plan to study nonlinear topological systems with chiral symmetry only.

Time-reversal symmetry does not serve to quantize nonlinear Berry phase. It demonstrates the real-valuedness of nonlinear bulk mode frequencies. In fact, numerical shooting method and analytic method of multiple-scale already manifest real frequencies of nonlinear bulk modes. We use time-reversal symmetry to mathematically confirm this result.

Summary of the modifications: We thank the referee for asking this question. In SI.II.B, Page.5, Paragraphs.2 and 3 marked by blue, we state how reflection symmetry constructs the relationships between the nonlinear mode components and quantizes nonlinear Berry phase. We add a new paragraph in the main text Sec.II, Page.4, right

column, Paragraph.2 marked by blue, to discuss the role of chiral symmetry in strongly nonlinear systems. The details of chiral symmetry are addressed in SI.II.G, Pages.10 and 11 for symmetry analysis, and in SI.IV.D, Pages.22 and 23 for edge mode derivations. Finally, we raise additional discussions in SI.II.A, Page.4, left column, Paragraph.2, marked by blue, to state the interplay between time-reversal symmetry and the real-valuedness of nonlinear mode frequencies.

“10) The authors should show the bulk-edge correspondence in strong nonlinear systems. What is the definition of the topological edge states in strong nonlinear systems? Is it possible to derive the nonlinear topological edge states analytically?”

We agree with the referee that bulk-boundary correspondence is important for the topological characterization, and nonlinear topological modes can be derived analytically. Based on Eq.(7) of the main text, we doubly confirm this correspondence using numerical computation and analytical calculation, to obtain nonlinear topological modes associated with the non-trivial quantized Berry phase. We then discuss the definition of nonlinear topological edge states.

In the **answer to Question #3**, the topological number picks the non-trivial value $\gamma(A < A_c) = \pi$ and the trivial value $\gamma(A > A_c) = 0$. In Figs.2(a) and (e) of the main text, we numerically verify bulk-boundary correspondence by imposing Gaussian shaking signals on the open boundary, and identify the presence and absence of nonlinear topological boundary mode below and above A_c in Figs.2(b,c) and (f,g), respectively. The corresponding numerical details are included in SI.IV.B, Page.20.

As shown in the main text Sec.III, Page.6, Eq.(8), the analytic strategy is to approximately solve nonlinear edge modes by truncating the motion equations to the fundamental harmonics. The analytic and numerical spatial profiles of mode amplitudes are depicted by the black dashed and red solid lines in Fig.2(d), respectively, and they are in good agreement with each other. Moreover, Eq.(8) has nonlinear boundary solutions only when $|\Psi_1^{(1)}| < A_c$, whereas no such solution exists for $|\Psi_1^{(1)}| > A_c$. These results are in good agreement with the non-trivial ($\gamma(A < A_c) = \pi$) and trivial ($\gamma(A > A_c) = 0$) topological invariants, which is the manifestation of bulk-boundary correspondence in the strongly nonlinear regime. Finally, the analytic details of nonlinear edge modes are carried out in SI.IV.B, Pages.18,19 from Eq.(4.12) to Eq.(4.16).

Next, we define strongly nonlinear topological boundary modes. To this end, we summarize their typical features. (1) Topological edge states cannot arise in systems under PBCs. (2) They emerge on lattice boundaries only when Berry phase picks the non-trivial quantized value, whereas they never exist for trivially quantized or un-quantized Berry phases. (3) Topological mode amplitudes are spatially non-uniform. (4) The frequencies stay in the band gap. (5) Topological modes are robust against disorders and randomness. In summary, we define nonlinear topological edge states as temporal-periodic nonlinear modes that satisfy the above five properties.

Summary of the modifications: The definition of nonlinear topological boundary modes is now incorporated in SI.IV.E, Page.23, marked by blue. By raising the above-mentioned five properties, we define topological boundary modes in the nonlinear regime.

“11) The authors show a nonlinear band structure for bulk. Is it possible to show a nonlinear band structure for a finite chain, where the nonlinear edge states are manifest?”

We thank the referee for raising this interesting question. We have been thinking about the nonlinear band structures for a finite chain under open boundary conditions (OBCs). The idea certainly works for linear systems where both bulk and topological edge mode frequencies are manifested. However, this idea cannot be extended to strongly nonlinear systems, because nonlinear modes that yield OBCs cannot be the linear superposition of nonlinear bulk modes under PBCs.

In linear systems, bulk modes under PBCs yield the format $e^{iqn-i\omega t}$, whereas standing waves read $\sin qn \sin \omega t$ to satisfy the vanishing requirements of OBCs. Since standing waves and bulk states are linear superpositions of one another, the band structures for PBCs and OBCs are almost identical except for the addition of localized mode frequencies. However, linear superposition is not valid for nonlinear modes. The amplitudes of a nonlinear bulk mode are spatially uniform under PBCs, whereas nonlinear modes yields vanishing conditions at open boundaries. These modes are not convertible to each other. Consequently, mode frequencies under OBCs differ from those under PBCs.

Summary of the modifications: We have incorporated the above discussions in SI.III.B, Page.15, marked by blue.

“12) Is it possible to characterize a self-induced transition using the topological number presented in this paper, which is a typical phenomenon in nonlinear systems? See Refs.26 and 44.”

We agree with the referee’s comment that self-induced topological transition and states can indeed be characterized by the topological number presented in this paper. It is worth emphasizing that Refs.[8,19] (i.e., Refs.[26] and [44] of the old manuscript) study self-induced transition in the weakly nonlinear regime by enabling perturbation theory. Below we show self-induced topological transition in the strongly nonlinear regime. We use two parallel methods, namely numerical computation and analytic derivation, to doubly confirm this transition.

According to Refs.[8,19], self-induced transition emerges in the weakly nonlinear regime if the lattice experiences “non-topological phase to topological phase transition” as the mode amplitudes rise. Therefore, we consider the same topological transition in the strongly nonlinear system. Quantized nonlinear Berry phase now picks the trivial value $\gamma(A < A_c) = 0$, and the non-trivial value $\gamma(A > A_c) = \pi$.

Bulk-boundary correspondence and the topological number together predict that the lattice is free of topological edge modes for $A < A_c$, whereas self-induced topological modes should emerge for $A > A_c$. The absence and presence of self-induced topological modes are numerically identified in the main text Sec.III, Fig.3(b,c) and (f,g) for $A < A_c$ and $A > A_c$, respectively.

In the analytical perspective, we conduct Eq.(8) of the main text. The semi-infinite lattice hosts topological evanescent modes when $|\Psi_1^{(1)}| > A_c$, whereas no such mode exists for $|\Psi_1^{(1)}| < A_c$. These results of self-induced topological modes are characterized by the topological number presented in this paper.

Summary of the modifications: We thank the referee for asking this important question. We discussed the anomalous behaviors of self-induced topological modes in Sec.III, Page.7, left column, Paragraph.1 of the main text and we regret that we were unclear. We have now addressed additional information in this paragraph marked by blue: “The anomalous behaviors of topological edge states are analogous to those in Refs.[7,26], in which self-induced topological transition is derived in beyond-Kerr weakly nonlinear metamaterials by enabling perturbation theory. Here, the self-induced topological phase is extended to the strongly nonlinear regime and is precisely characterized by the topological number in Eq.(3).” The corresponding analytic and approximate derivation is carried out in SI.IV.B, Page.18,19 by truncating the motion equations to the fundamental harmonics to obtain the spatial recursion relation of nonlinear mode amplitudes.

“13) It is not clear why the electrical circuit shown in Fig.5 is nonlinear because all elements are linear. The derivation of Eq.(5.11) in SM is also not clear. The authors should add more detailed derivations.”

FIG. 2. Experimental proposals of topological electrical circuits. (a) Unit cell of purely linear topoelectrical circuit without nonlinear elements. (b) Unit cell of nonlinear topoelectrical circuit by introducing active voltage sources $\delta V_n^{(1)}(V_n^{(2)}, V_{n-1}^{(2)})$ and $\delta V_n^{(2)}(V_n^{(1)}, V_{n+1}^{(1)})$ nonlinearly controlled by voltage fields $V_n^{(j=1,2)}$.

We appreciate the referee’s remind that the current contents in the main text Fig.5 and SI.V are not clear enough. Below we take a two-step answer to address additional information. First, we discuss a linear topoelectrical circuit that hosts quantized Berry phase. Second, we add nonlinear electrical elements to the linear system and realize the

nonlinear topoelectrical circuit system.

First, we consider a linear topoelectrical circuit system in Fig.R2(a). The unit cell is composed of two *LCR* resonators with the natural frequency $\omega_0 = 1/\sqrt{LC}$, and the resonators are connected by linear capacitors C_1 and C_2 . Using Kirchhoff's law (SI.V Eqs.(5.1-5.3)), we obtain the motion equations of the ladder circuit in SI.V Eq.(5.5). Following the ansatz in Ref.[21], we express the state vector V that denotes the set of voltage fields as the envelope function $V = \Psi e^{-i\omega_0 t}$. In the undamped limit $RC\omega_0 \ll 1$, the motion equations are reduced to the standard Schrödinger equation $i\partial_t \Psi = H\Psi$, where H is the circuit Hamiltonian shown in SI.V Eq.(5.6), and the topological properties are ensured by reflection symmetry in H .

We now introduce nonlinear interactions to the circuit model by adding nonlinear voltage sources $\delta V_n^{(1)}$ and $\delta V_n^{(2)}$ in Fig.R2(b). These voltage sources are nonlinearly controlled by the voltage fields $V_n^{(1)}$ and $V_n^{(2)}$ to provide feedback to the motion equations. Nonlinear Berry phase is quantized if the interactions respect reflection symmetry. Thus, we demand that the voltage sources yield $\delta V_n^{(1)} = \delta V_n^{(1)}(V_n^{(2)}, V_{n-1}^{(2)})$ and $\delta V_n^{(2)} = \delta V_n^{(2)}(V_n^{(1)}, V_{n+1}^{(1)})$ to preserve reflection symmetry, but their functional forms can be arbitrary. Using the envelope function representation $\delta V = -\delta\Psi e^{-i\omega_0 t}$ for the state vector δV that denotes the set of nonlinear voltage sources $\delta V_n^{(1)}$ and $\delta V_n^{(2)}$, the lattice dynamics is now depicted by the generalized nonlinear Schrödinger equation

$$i\partial_t \Psi = H\Psi + \delta\Psi, \quad (6)$$

where $\delta\Psi$ yields reflection symmetry and is nonlinearly controlled by $\Psi = V e^{i\omega_0 t}$. Finally, to realize the model (7) of the main text, we ask that the nonlinear voltage sources satisfy $\delta\Psi_{2n-1} = f_1(\Psi_{2n-1}, \Psi_{2n}) + f_2(\Psi_{2n-1}, \Psi_{2n-2})$ and $\delta\Psi_{2n} = f_1(\Psi_{2n}, \Psi_{2n-1}) + f_2(\Psi_{2n}, \Psi_{2n+1})$.

In our recent mechanical experiment[22] (Fig.R4 of the **answer to Referee 3, Question #4** below), the interaction in Eq.(7) of the main text is also realized in the 1D mechanically isostatic metamaterial which is composed of elliptic gears. The nonlinear extension of topological floppy modes, states of self-stress, and topological robustness have been experimentally confirmed in this mechanically isostatic system.

Summary of the modifications: We appreciate the referee's important remind. We rewrite the main text, Sec.IV, Page.7, right column, Paragraphs.1 and 2, where the new contents are marked by blue. By connecting to the existing literature[21], we introduce nonlinear voltage feedback to the linear circuit and obtain the nonlinear topological circuit model. We make revisions in SI.V, Pages.24 and 25, marked by blue, to address the corresponding mathematical derivations.

II. REPLY TO REFEREE 2

“The manuscript describes topological aspects of a nonlinear Schrödinger equation in a two-component system. The work builds on a recent line of development that has found interest in many areas of physics and engineering, and is very interesting. It appears to have been carried out carefully, and the results are all sound.”

We appreciate the positive recommendation of the referee. Below we respond to the two specific questions (marked by Questions #1 and #2) raised by the referee. We use “Eq.R(X) and Fig.R(X)” to refer to the equations and figures in this reply, and use “Eq.(X) and Fig.(X)” to refer to the equations and figures that appear in the citing references. The referee's questions and comments are quoted in **blue**, with our responses given in **black** and any text quoting or referencing relevant changes we've made given in **bold**. At the end of each answer, we summarize the modifications in the revised manuscript.

“1) I only have a comment regarding the stability of the presented topological features with respect to the wave-like nature of the bulk solution. The justification used by the authors to assume the pure wavelike form for the bulk solution is based on their computational observations in a number of models, presented in the supplement. However, some of those results show significant additional components to the pure wave. What would be interesting to see is a perturbation analysis of the nonlinear equation around the wavelike solution, and an analysis of the stability of the topological features with respect to the new linear contributions. Can this stability analysis also leads to new information about the symmetry requirements? Such an additional analysis will enhance the work of the authors significantly.”

We appreciate the referee for asking this important question. Stability plays an essential role for the validity of nonlinear modes. Inspired by the referee's question, we indeed observe new and exciting results: The intrinsic symmetries of the nonlinear system guarantee that all nonlinear bulk modes are *marginally stable* within Floquet analysis. Below, we analytically study (1) the stability of nonlinear bulk modes, (2) the stability of the topological index, and (3) the stability of nonlinear topological boundary modes.

(1) First, we conduct the interplay between Floquet analysis and system symmetries to demonstrate the marginal stability of nonlinear bulk modes. Numerical computation is further performed to confirm this marginal stability.

We perturb the nonlinear bulk mode $\Psi_{(0)}$ with a small wave disturbance $\delta\Psi$. The wave equation for $\delta\Psi$ reads

$$\partial_t \delta\Psi = M(\Psi_{(0)})\delta\Psi + \mathcal{O}(\delta\Psi^2), \quad (7)$$

where the monodromy matrix $M(\Psi_{(0)})$ is periodically modulated by $\Psi_{(0)}$ with the period $T = 2\pi/\omega$, and $\mathcal{O}(\delta\Psi^2)$ stands for higher-order terms. In the small-disturbance limit, we linearize Eq.R(7) by dropping higher-order terms. Hence, the stability analysis of nonlinear bulk modes is mathematically formulated as a Floquet problem. The stability properties are captured by the Floquet exponents[23-25] $\{\alpha_n\}$ determined by the eigenvalues of $\tilde{M} = \int_0^T M(\Psi_{(0)}(t))dt$, where \tilde{M} is dubbed the time-domain monodromy matrix. As indicated by supplementary information (abbreviated as ‘‘SI’’ in the rest of this reply), Sec.II.E, Eqs.(2.29) and (2.30), the intrinsic symmetries of the nonlinear model guarantee that \tilde{M} is always real and *anti-symmetric*. Consequently, all Floquet exponents are purely imaginary, and the nonlinear bulk mode is marginally stable.

FIG. 3. Statistics of $\text{Re } \alpha$ for 500 randomly generated disturbing waves versus the mode amplitudes. (a) The curve centers are mean values of $\text{Re } \alpha$, and the error bars are the corresponding standard deviations. (b) Enlarged window in (a) with amplitudes ranging from $A = 0.7$ to 1.13 .

However, higher order terms in Eq.R(7) slowly accumulate over time and may be non-negligible as $\delta\Psi$ grow large. As detailed in SI.II.E Pages.9 and 10, we take one step further to investigate the mode stability by the following numerical strategy. The algorithm is to randomly generate 500 wave disturbances and record the Floquet exponents of the perturbed modes over 100 periods of free oscillations. For each wave disturbance (i.e., for every perturbed mode), we average over the biggest real part of the Floquet exponents for 100 periods, and denote it as $\text{Re } \alpha$. We perform the statistical average and standard deviation for 500 different $\text{Re } \alpha$. In Figs.R3, we plot the average and standard deviation of $\text{Re } \alpha$ versus the amplitudes of nonlinear bulk modes.

For amplitudes $A \lesssim 0.9$, $\text{Re } \alpha$ fluctuates within $\pm 10^{-4}$ for 100 periods and is in line with the marginal stability derived from analytic Floquet analysis. For $1.1 \gtrsim A \gtrsim 0.9$, modes manifest weak instability due to the accumulated effect of higher-order wave disturbances. As amplitudes grow beyond $A \gtrsim 1.1$, modes become more unstable due to the increase in the Floquet exponents. As the referee points out, the instability of nonlinear modes is also reflected by the additional components to the pure wave, such as the nonlinear modes displayed in SI.Fig.5(c,e).

(2) Second, we consider the stability of the quantized Berry phase γ with respect to mode disturbances. We note that geometric Berry phase γ is defined from *nonlinear bulk modes*. Thus, the perturbed wave has to be a nonlinear bulk mode in order to have a well-defined γ , otherwise the wave number is ill-defined, and the adiabatic evolution of the wave number cannot be performed. The motion equations demand that the perturbed nonlinear bulk mode still respects reflection symmetry. As shown in Eq.(6) of the main text, the perturbed nonlinear bulk mode also guarantees the integer-value (i.e., the stability) of nonlinear Berry phase γ , as long as the disturbance is not large enough to cause topological phase transition.

(3) Finally, we show that nonlinear topological boundary modes are stable. In contrast to spatially homogeneous amplitudes of nonlinear bulk modes, the spatial profile of boundary modes is inhomogeneous and breaks the crystalline symmetry. The Floquet monodromy matrix lacks of anti-symmetry, and the properties of Floquet multipliers are unknown.

Nevertheless, the stability of nonlinear topological modes can be studied in a perturbative manner using the *method of multiple-scale* in the weakly nonlinear regime. To enable the stability analysis, we introduce a small damping in the equations of motion, as shown in SI.IV.A Eqs.(4.2). Method of multiple-scale introduces a small book-keeping parameter $\epsilon \ll 1$ on the damping and the nonlinear coefficients to enforce small amplitudes. This method further expands the time derivative in orders of slow-time variables, as detailed in SI.III.A Eqs.(3.2) and (3.3), and in SI.IV.A. Thus, the temporal evolution of mode amplitudes is expressed as

$$\partial_t A = D_0 A + \epsilon D_1 A + \epsilon^2 D_2 A + \dots, \quad (8)$$

where D_0 , D_1 and D_2 are zeroth, first, and second-order time derivatives, respectively. The mode is stable (unstable) when $\partial_t A < 0$ ($\partial_t A > 0$). Using method of multiple-scale in SI.IV.A, we determine $\partial_t A$ by solving $D_0 A$ (Eq.(4.4)), $D_1 A$ (Eqs.(4.6), (4.7)), and $D_2 A$ (Eqs.(4.8)~(4.10)). Consequently, nonlinear topological modes are stable (unstable) when the amplitudes yield $A < A_{c,\text{stability}}$ ($A > A_{c,\text{stability}}$), where $A_{c,\text{stability}}$ is the critical amplitude, above which nonlinear topological edge modes lose stability. We note that in the weakly nonlinear regime, $A_{c,\text{stability}}$ and the topological transition amplitude are equal, meaning that all weakly nonlinear topological boundary modes are stable.

Summary of the modifications: We thank the referee for raising this important question. We make a number of changes in the main text and supplementary information.

The stability of nonlinear bulk modes is updated in the revised main text, Sec.II, Page.3, left column, Paragraph.1 marked by blue: “Due to the symmetry constraints of the nonlinear motion equations, we demonstrate that all nonlinear bulk states are marginally stable within Floquet analysis (see SI.II.E for details).” The corresponding detailed discussions are incorporated in SI.II.E, Pages.8, 9, 10 marked by blue, where we employ Floquet equation and symmetry analysis to characterize the stability of nonlinear bulk modes in analytic and numerical ways.

The stability of the topological index γ is now added in the main text, Sec.II, Page.4, left column, Paragraph.3, marked by blue: “In the strongly nonlinear regime, γ still manifests stability against mode disturbances by staying as the integer. Corresponding demonstrations are carried out in SI.II.B.” In SI.II.B, Page.5, right column, Paragraph.3, we incorporate additional discussions marked by blue by stating that the quantized γ stays invariant because the perturbed nonlinear bulk mode still respects reflection symmetry.

The stability of nonlinear topological modes is incorporated in the main text, Sec.III, Page.6, right column, Paragraph.1 marked by blue: “These nonlinear topological states are stable against mode disturbances, which is mathematically demonstrated in SI.IV.A.” The technical details are addressed in SI.IV.A, Pages.17, 18, Eqs.(4.7), (4.10) and (4.11) using method of multiple-scale.

“2) My other query will be with regards to the choice of 1D. How robust is the behavior reported by the authors with respect to the choice of dimensionality? For example, can they comment on what would happen if they considered a 2D system, such as the stochastic model studied in Tang et al, PRX 11, 031015 (2021)?”

We agree with the referee that our results are robust with respect to the choice of dimensionality. We also appreciate the referee for bringing this interesting work[26] to our attention: Tang et. al., Phys. Rev. X **11**, 031015 (2021). This paper is indeed much related to our work because nonlinear interactions beyond Kerr-type are ubiquitous in biological processes. However, there are difficulties to directly apply our results in biological systems.

First, we show that nonlinear topological invariant and phases in 1D can be extended to higher-dimensional systems. In linear systems, quantized Berry phase, topological boundary states, and Weyl bulk modes are typical topological properties of two-dimensional and higher-dimensional systems. Likewise, recently we manage to demonstrate these features in strongly nonlinear 2D systems[5], where nonlinear Weyl modes are topologically protected by quantized nonlinear Berry phase along trajectories enclosing Weyl singularities, and these singularities are the degenerate points of nonlinear band structures. For topologically polarized lattices free of Weyl points, nonlinear topological modes are manifested on lattice open boundaries. Based on the preliminary results of nonlinear Weyl modes[5] in 2D systems, we agree with the referee’s intuition that the results in the current work should be extendable to 3D and higher-dimensional systems. Thus, nonlinear topological band theory is robust with respect to the choice of dimensionality.

Nevertheless, a number of problems emerge if we directly apply our work to study nonlinear stochastic bio-physical processes. In the work[26] by Tang et. al. (Phys. Rev. X **11**, 031015 (2021)), the authors conducted topological

properties in a linear non-equilibrium stochastic process, where bulk mode eigenfrequencies are complex in the non-Hermitian system. Complex eigenfrequencies imply that the mode amplitudes exponentially shrink or grow over time. In nonlinear systems like the stochastic biological process, complex frequencies also lead to the temporal change of mode amplitudes, but increasing nonlinear interactions provide feedback and modify the frequencies. Consequently, frequencies do not stay constant in time, which are in sharp contrast to the constant complex eigenfrequencies of linear systems. Thus, a systematic investigation of complex frequencies in nonlinear dynamics is required before we further study their topological attributes.

Another problem of complex frequencies in nonlinear dynamics is associated with the celebrated “non-Hermitian skin effect” [30,31]. In linear systems, complex eigenfrequencies of bulk modes under periodic boundary conditions (PBC) indicate that bulk modes under open boundary conditions (OBCs) are no longer plane-wave-like states. Instead, bulk modes under OBCs exponentially localize on lattice boundaries and largely modify boundary responses. This is known as the “non-Hermitian skin effect”. In nonlinear systems, it is natural to expect that complex frequencies under PBCs also ensure the localized behavior of nonlinear modes under OBCs. However, unlike linear systems, the spatially inhomogeneous nonlinearities may provide feedback and modify the frequencies. It is therefore intriguing to conduct new physics about the boundary responses in stochastic nonlinear dynamics.

Following this interesting work [26], we agree with the referee that the interplay between nonlinear topological physics and stochastic biological processes is an exciting and challenging field that is worth exploration.

Summary of the modifications: We thank the referee for introducing this important work [26]. In the revised main text, we add a new section titled “summary and perspectives” in Sec.V, Page.8, marked by blue: Together with the **answer to Question #1 of Referee 1**, we discuss prospective interesting directions of nonlinear topological physics inspired by Refs.[1,2,26].

III. REPLY TO REFEREE 3

“In this review, I am commenting on the submitted article ”Topological Invariant and Anomalous Edge States of Strongly Nonlinear Systems” by Zhou et al. Unfortunately, I will keep my analysis of the issues short in the hopes that the authors may reconsider the form and structure of their presentation. Simply put, I do not find the work interesting, at all. There is not enough context for one who to be able to identify the reason that this work is interesting far less understand how it represents an advance in the field that changes the manner in which I view the field of topology. The work begins in a highly mathematical manner and it continues to be so for the duration of the paper providing little insight into approximations or limitations. Very early in the manuscript, there is a mention that the work will not address the turbulent regime, but it is not really clear what regime is being addressed here/ The work does not, until in a very passing manner, describe anything that resembles a physical system and so it is not possible to make the connection between the physical nature of a system and the mathematics that is presented. As the presented work touches on circuit models, then, presumably, there would have been an easy connection with the significant literature on topological circuits that currently exists. Furthermore, from a purely practical perspective, the interest in this work, as currently written, is very clearly limited to those who are theoretically inclined as I cannot see how an experimentalist is going to wade through this math.

In the end, maybe there is something interesting here but I certainly cannot see it. The manuscript does not change the manner in which I view the field of topology broadly nor does it give me additional insights into a system that has been the canonical workhorse for new models. I cannot recommend publishing this work.”

We appreciate the constructive criticisms of the referee. Below we respond to the specific criticisms/questions (marked by Questions #1~#4) raised from the referee. The referee’s questions and comments are quoted in **blue**, with our responses given in **black** and any text quoting or referencing relevant changes we’ve made given in **bold**. At the end of each answer, we summarize the modifications in the revised manuscript.

“1) Unfortunately, I will keep my analysis of the issues short in the hopes that the authors may reconsider the form and structure of their presentation. Simply put, I do not find the work interesting, at all. There is not enough context for one who to be able to identify the reason that this work is interesting far less understand how it represents an advance in the field that changes the manner in which I view the field of topology. The work begins in a highly mathematical manner and it continues to be so for the duration of the paper providing little insight into approximations or limitations.”

We thank the referee for raising this criticism. First of all, we agree with the referee’s point of view that restructuring our manuscript, such as moving the circuit example to the beginning, certainly improves the accessibility. However, moving the circuit may cause problems. For example, the readers may have the impression that our theory only works for circuits, but as we show in the **answer to Question #4**, Fig.R4, our theory has been experimentally demonstrated in nonlinear topological isostatic metamaterials[22]. Referee 2 points out that our work also applies to stochastic bio-physical cycles[26]. Re-structuring the manuscript may change the emphasis to a specific example and obscure the generality of our theory. Finally, Referees 1 and 2 are fine with the current structure that we present the generality of the theory first, and then propose a physical realization as one of the examples. Therefore, we respectfully request the referee that we revise the content by keeping the current logic.

We now state how our work advances the field of nonlinear topology. To this end, we introduce existing experiments. Recent advances[33-36] in nonlinear topological *photonics* are studied by the topological index exactly derived from Kerr-nonlinearities. Due to the particularity of Kerr-nonlinear interactions, the Kerr topological invariants are the same as those in linear topological theory. Another class is the *weakly* nonlinear “beyond-Kerr” topological systems[7,8,18,21,37], where the interactions are transformed to Kerr-nonlinearities by dropping higher-order harmonics, and the invariants are Kerr topological indices as well. However, the nice agreement between theory and experiments cannot be carried over to systems that are neither Kerr nor weakly nonlinear.

We then discuss *strongly* nonlinear “beyond-Kerr” systems, where topological indices are hardly discussed due to the mathematical challenge. This seems to merely be a theoretical problem. However, the growing literature[1,2,18,38-41] begin to aware that as amplitudes grow, Kerr topological invariants are not always correct to predict the topological properties in experiments due to three main reasons. (1) Nonlinear edge modes[21] are no longer topological for large enough amplitudes because the frequencies are shifted by *on-site* nonlinearities and merge with bulk bands[1,2,40]. (2) Higher-order harmonics are non-negligible because they modify the topological transition amplitude. The topologically trivial system can be misleadingly predicted as “non-trivial” by neglecting higher-order harmonics in Kerr topological index. (3) Symmetry-breaking higher-order harmonics may destroy the topological nature[1,2] of the nonlinear system.

Due to the lack of theoretical foundations, current topological experiments are restricted to Kerr-nonlinear photonics[33-36], and weakly nonlinear systems[7,8,18,21,37]. The ubiquitous strongly nonlinear “beyond-Kerr” systems necessitate a new index to characterize their topology. In this manuscript, we address this shortcoming by introducing the nonlinear topological index that serves as the step stone of future strongly nonlinear topological experiments.

There may be new physics in “beyond-Kerr” topological systems, as pointed out by **Question #7 of Referee 1**. In contrast to Kerr-interactions that restrict *equal* topological transition amplitudes $A_c(\text{real}) = A_c(\text{imaginary})$ for real and imaginary parts of field variables, “beyond-Kerr” nonlinearities can break this constraint. It is intriguing to ask what happens in the intermediate amplitude regime between $A_c(\text{real})$ and $A_c(\text{imaginary})$. As pointed out by **Question #1 of Referee 2**, reflection symmetry and lattice topology demand that all nonlinear bulk modes are *marginally stable*. Therefore, it is interesting to further explore how symmetry properties control nonlinear dynamics.

Summary of the modifications: We appreciate the referee for raising this criticism. In the revised main text, we make a number of changes marked by blue.

The universality and wide applicability of our theory are emphasized in the introduction, Sec.I, Page.1, right column, Paragraph.2: “In spite of the remarkable different physical origins of mechanical isostatic structures, electrical circuits, deep water waves, and bio-physical cycles, their dynamics are commonly described by generalized nonlinear Schrödinger equations, which we adopt to study theoretically, for future nonlinear topological experiments.” In Sec.II, Page.2, left column, Paragraph.1: “In Sec.IV, we propose an electrical circuit as one of the classical systems that yield Eqs.(1). Following our work, recent experiment conducts an isostatic metamaterial that manifests strongly nonlinear boundary floppy modes. Their topological nature is also captured by Eqs.(1).”

The problems of using Kerr topological index for strongly nonlinear “beyond-Kerr” systems are discussed in the revised main text, Sec.I, Page.1, left column, Paragraph.3, marked by blue: (1)“Though boundary modes remain topologically protected in the weakly nonlinear regime, strong nonlinearities may destroy their topological nature by breaking the intrinsic symmetries, and existing linear and weakly nonlinear topological theories are not always correct to predict their strongly nonlinear topological properties;” (2)“Thus, it is demanding to invoke the topological number that precisely describes the topological attributes of beyond-Kerr strongly nonlinear systems.”

The future directions of nonlinear topological physics are addressed in the revised main text, Sec.V titled “Summary and perspectives”, Page.8, marked by blue. Together with the **answer to Question #1 of Referee 1**, and the **answer to Question #2 of Referee 2**, we discuss interesting directions of nonlinear topological physics inspired by Refs.[1,2,26].

“2) Very early in the manuscript, there is a mention that the work will not address the turbulent regime, but it is not really clear what regime is being addressed here.

We appreciate the referee for bringing this important question. Turbulence is associated with the instability of nonlinear modes. As stated in the **answer to Question #1 of Referee 2**, the intrinsic symmetries of the topological system guarantee that all nonlinear bulk modes are *marginally stable*. As shown in Fig.R3, for amplitudes $A \lesssim 0.9$, the nonlinear lattice is stable against disturbances and turbulence will not emerge. For $0.9 \lesssim A \lesssim 1.1$, Floquet analysis indicates weak instability because mode disturbances increase by $\sim 10\%$ after 10 periods. According to Refs.[12,15], the nonlinear modes do not fall apart within 10 periods, and turbulence does not occur in this amplitude regime. Moreover, friction is ubiquitous in classical systems. Small damping[3,22] can balance the weak instability in nonlinear modes and stabilize the system. Therefore, turbulence will not occur within the interested amplitude scope $A \leq 1.1$ of this work.

Summary of the modifications: We thank the referee for pointing out this important question and regret that we were unclear. The stability of nonlinear bulk modes is updated in the revised main text, Sec.II, Page.3, left column, Paragraph.1, marked by blue.: “Due to the symmetry constraints of the nonlinear motion equations, we demonstrate that all nonlinear bulk states are marginally stable within Floquet analysis (see SI.II.E for details).” Corresponding details are addressed in the supplementary information (abbreviated as “SI” in the rest of this reply) II.E, Pages.8, 9, 10 marked by blue, where both analytic and numerical methods are employed to elaborate on the marginal stability of nonlinear modes.

“3) The work does not, until in a very passing manner, describe anything that resembles a physical system and so it is not possible to make the connection between the physical nature of a system and the mathematics that is presented. As the presented work touches on circuit models, then, presumably, there would have been an easy connection with the significant literature on topological circuits that currently exists.

We agree with the referee’s constructive suggestion that additional information should be addressed on the physical systems. Therefore, we make connections with existing literature of topological circuits by tracing back to Ref.[21], where the circuit model presents topological boundary modes in the *weakly nonlinear regime*. Recent works[1,2,40] point out that on-site nonlinearities induced by the nonlinear capacitors[21] can shift the frequencies of topological boundary modes. The frequency shift is negligible in weakly nonlinear systems[21], but can be large in the strongly nonlinear regime such that edge mode frequencies merge into bulk bands. Strongly nonlinear boundary modes become blur[1] and are no longer topologically protected[2,40]. To avoid the frequency shift, we eliminate on-site nonlinearities by replacing nonlinear capacitors in Ref.[21] with linear ones. In our model, the nonlinearities are introduced by active voltage sources connected to the *LCR* resonators, as shown by $\delta V^{(1)}$ and $\delta V^{(2)}$ in Fig.R2(b) and Fig.5 of the main text. The dynamics is clear of *on-site* nonlinearities, and the frequencies of nonlinear topological modes stay in the nonlinear band gap.

Summary of the modifications: We make connections to existing literature in Sec.I of the main text, Page.1, right column, Paragraph.2 marked by blue. As stated in the **answer to Question #1**, we revise the main text in Sec.II, Page.2, left column, Paragraph.1 by connecting Eqs.(1) of the main text to the circuit model and the mechanically isostatic experiment[22].

The poeoelectrical circuit has been re-written in the main text Sec.IV, Page.7, right column marked by blue. We build our circuit model by comparing to Ref.[21]. In SI.V, Pages.24, 25 marked by blue, we address the details of the poeoelectrical circuit by connecting to Ref.[21].

“4) Furthermore, from a purely practical perspective, the interest in this work, as currently written, is very clearly limited to those who are theoretically inclined as I cannot see how an experimentalist is going to wade through this math.

We thank the referee for raising this criticism. This work aims to solve the open question in topological experiments[21,22,38,41] that the breakage of existing theory-experiment correspondence necessitates a new topological index for strongly nonlinear systems. We regret that our presentation has made the referee feel as though a reader would have to “wade” through all of the math to derive any benefits from our results. We do not wish our paper to necessarily be read in this manner, any more than an experimentalist would wish their papers to be read only by those who understood all of the details of experimental methodology. We do regard certain mathematical aspects to be essential for analytic completeness and rigorousness: the equations of motion, the wave solutions and the topological invariants. Other expressions in the main text only serve to create a coherent narrative flow and need not be worked

FIG. 4. (a) Elliptic gears are assembled to a 1D mechanically isostatic chain[22]. (b) The lattice hosts strongly nonlinear topological floppy modes localized on the left open boundary.

through carefully by a reader interested only in results.

To describe our results non-mathematically, we employ figures in standard formats, such as quantized topological number (Fig.1(d)), nonlinear dispersion relations (Fig.1(e)) and spatial-temporal plots of intensity (Fig.2(b),(f), Fig.3(b),(f), and Fig.4(b),(f)). We also make connections to past experiments in Sec.I, II, IV, and V. The purposes and details of making these revisions are itemized in the **answers to Questions #1, 2, 3** above.

Then, in order to design an experiment, one only needs to focus on the conclusion outlined in the abstract and introduction: Strongly nonlinear boundary modes are topologically protected as long as symmetries are respected. Other mathematical details are unnecessary for experimental designs, as we shall illustrate in the following nonlinear mechanically isostatic experiment[22].

We hope that this approach has preserved the novelty of our work while still making it accessible to as broad an audience as possible.

We now discuss the experiment[22] that extends the notions of topological isostatic lattices[31,38,39,42], floppy modes and states of self-stress, to the strongly nonlinear regime. The characterization of topological phases necessitates the nonlinear topological index presented in this work. As shown in Fig.R4, the system consists of an elliptically geared mechanical chain subjected to OBCs. Gears are freely rotatable about fixed pivots on the right-sided focal points. Nearest-neighbor gears keep engaged during rotations. The sliding distances between touching gears $\ell_n = \ell_n(\theta_n, \theta_{n+1})$ are nonlinearly controlled by rotation angles. In the linear regime when $\theta/\pi \ll 1$, the sliding distances are depicted by the compatibility matrix[42] $C(q)$ that manifests quantized Berry phase and topological boundary floppy modes. In the strongly nonlinear regime for $\theta/\pi \sim \mathcal{O}(1)$, boundary floppy modes are experimentally confirmed[22], as shown in Fig.R4(b). The topological protection of the boundary modes is assured by the strongly nonlinear topological index, and is reflected by the mode robustness against disorders in gears.

It is notable that elliptic geometry is not special to realize nonlinear topological floppy modes. These topological modes manifest on system boundaries as long as the intrinsic symmetries are preserved, regardless of the choice of gear geometry. As illustrated in Ref.[22], nonlinear topological floppy modes also emerge in a chain of gears consisting of conjugated triangles and trefoils.

REFERENCES

- [1] R. W. Bomantara, W. Zhao, L. Zhou, and J. Gong, Nonlinear Dirac cones, Phys. Rev. B **96**, 121406(R) (2017).
- [2] T. Tuloup, R. W. Bomantara, C. H. Lee, and J. Gong, Nonlinearity induced topological physics in momentum space and real space, Phys. Rev. B **102**, 115411 (2020).
- [3] D. Zhou, D. Z. Rocklin, M. Leamy, and Y. Yao, Topological invariant and anomalous edge states of strongly nonlinear systems, revised manuscript resubmitted for review.
- [4] F. D. M. Haldane, Model for a quantum Hall effect without Landau levels: Condensed-matter realization of the parity anomaly, Phys. Rev. Lett., **61**, 2015 (1988).
- [5] D. Zhou, J. Zhang, and Y. Yao, Weyl modes in strongly nonlinear systems, working in progress.

- [6] S. Ryu, A. P. Schnyder, A. Furusaki, and A. W. W. Ludwig, Topological insulators and superconductors: tenfold way and dimensional hierarchy, *New J. Phys.* **12** 065010 (2010).
- [7] Y. Hadad, A. B. Khanikaev, and A. Alu, Self-induced topological transitions and edge states supported by nonlinear staggered potentials, *Phys. Rev. B* **93**, 155112 (2016).
- [8] R. Chaunsali, and G. Theocharis, Self-induced topological transition in phononic crystals by nonlinearity management, *Phys. Rev.* **100**, 014302 (2019).
- [9] X. Zhou, Y. Wang, D. Leykam and Y. D. Chong, Optical isolation with nonlinear topological photonics, *New J. Phys.* **19** (2017) 095002.
- [10] L. Renson, G. Kerschen, and B. Cochelin, Numerical computation of nonlinear normal modes in mechanical engineering, *Journal of Sound and Vibration* **364**, 177 (2016).
- [11] S. N. Ha, A nonlinear shooting method for two-point boundary value problems, *Computers & Mathematics with Applications* **42**, 1411 (2001).
- [12] M. Peeters, R. Viguie, G. Serandour, G. Kerschen, and J.-C. Golinval, Nonlinear normal modes, Part II: Practical computation using numerical continuation techniques, in 26th International Modal Analysis Conference, Orlando, 2008 (2008).
- [13] R. K. Narisetti, M. J. Leamy, and M. Ruzzene, A Perturbation Approach for Predicting Wave Propagation in One-Dimensional Nonlinear Periodic Structures, *Journal of Vibration and Acoustics* **132** (2010).
- [14] R. Zaera, J. Vila, J. Fernandez-Saez, and M. Ruzzene, Propagation of solitons in a two-dimensional nonlinear square lattice, *International Journal of Non-Linear Mechanics* **106**, 188 (2018).
- [15] M. D. Fronk and M. J. Leamy, Higher-Order Dispersion, Stability, and Waveform Invariance in Nonlinear Monoatomic and Diatomic Systems, *Journal of Vibration and Acoustics* **139** (2017).
- [16] D. D. Snee and Y.-P. Ma, Edge solitons in a nonlinear mechanical topological insulator, *Extreme Mechanics Letters* **30**, 100487 (2019).
- [17] W. P. Su, J. R. Schrieffer, and A. J. Heeger, Solitons in Polyacetylene, *Phys. Rev. Lett.* **42**, 1698 (1979).
- [18] R. Chaunsali, H. Xu, J. Yang, P. G. Kevrekidis, and G. Theocharis, Stability of topological edge states under strong nonlinear effects, *Phys. Rev. B* **103**, 024106 (2021).
- [19] Y. Hadad, A. B. Khanikaev, and A. Alu, Self-induced topological transitions and edge states supported by nonlinear staggered potentials, *Phys. Rev. B* **93**, 155112 (2016).
- [20] J. Tang, F. Ma, D. Zhou, Y. Yao, and F. Li, Strongly nonlinear topoelectrical circuits, working in progress.
- [21] Y. Hadad, J. C. Soric, A. B. Khanikaev, and A. Alu, Self-induced topological protection in nonlinear circuit arrays, *Nature Electronics* **1**, 178 (2018).
- [22] D. Zhou, W. Ying, X. Shi, Z. Tang, J. Yang, F. Li, and Y. Yao, Strongly nonlinear transitions of topological phases in elliptically geared chains, manuscript submitted.
- [23] M. Peeters, R. Viguie, G. Serandour, G. Kerschen, and J.C. Golinval, Nonlinear Normal Modes, Part II: Practical Computation using Numerical Continuation Techniques, in 26th International Modal Analysis Conference, Orlando, 2008 (2008).
- [24] A. Lazarus, O. Thomas. A harmonic-based method for computing the stability of periodic solutions of dynamical systems. *Comptes Rendus Mecanique*, Elsevier Masson, **338** (9), pp.510-517 (2010).
- [25] R. Kawaia, K. Lindenberg, and C. VandenBroeck, Parametrically modulated oscillator dimer: an analytic solution, *Physica A* **312** 119–140 (2002).
- [26] E. Tang, J. Agudo-Canalejo, and R. Golestanian, Topology Protects Chiral Edge Currents in Stochastic Systems, *Phys. Rev. X* **11**, 031015 (2021).
- [27] J. Knebel, P. M. Geiger, and E. Frey, Topological Phase Transition in Coupled Rock-Paper-Scissors Cycles, *Phys. Rev. Lett.* **125**, 258301 (2020).
- [28] T. Yoshida, T. Mizoguchi, and Y. Hatsugai, Chiral edge modes in evolutionary game theory: A kagome network of rock-paper-scissors cycles, *Phys. Rev. E* **104**, 025003 (2021).
- [29] T. Yoshida, T. Mizoguchi, and Y. Hatsugai, Non-Hermitian topology in evolutionary game theory: Exceptional points and skin effects in rock-paper-scissors cycles, arXiv:2109.11127.
- [30] S. Yao and Z. Wang, Edge States and Topological Invariants of Non-Hermitian Systems, *Phys. Rev. Lett.* **121**, 086803 (2018).
- [31] D. Zhou and J. Zhang, Non-Hermitian topological metamaterials with odd elasticity, *Phys. Rev. Research* **2**, 023173 (2020).
- [32] J. Claes and T. L. Hughes, Skin effect and winding number in disordered non-Hermitian systems, *Phys. Rev. B* **103**, L140201 (2021).
- [33] D. Smirnova, D. Leykam, Y. Chong, and Y. Kivshar, Nonlinear topological photonics, *Applied Physics Reviews* **7**, 021306 (2020).
- [34] L. J. Maczewsky, M. Heinrich, M. Kremer, S. K. Ivanov, M. Ehrhardt, F. Martinez, Y. V. Kartashov, V. V. Konotop, L. Torner, D. Bauer, et al., Nonlinearity-induced photonic topological insulator, *Science* **370**, 701 (2020).

- [35] S. Xia, D. Kaltsas, D. Song, I. Komis, J. Xu, A. Szameit, H. Buljan, K. G. Makris, and Z. Chen, Nonlinear tuning of PT symmetry and non-Hermitian topological states, *Science* **372**, 72 (2021).
- [36] X. Zhou, Y. Wang, D. Leykam, and Y. D. Chong, Optical isolation with nonlinear topological photonics, *New Journal of Physics* **19**, 095002 (2017).
- [37] R. K. Pal, J. Vila, M. Leamy, and M. Ruzzene, Amplitude-dependent topological edge states in nonlinear phononic lattices, *Physical Review E* **97** (3), 032209 (2018).
- [38] B. G. Chen, N. Upadhyaya, and V. Vitelli, Nonlinear conduction via solitons in a topological mechanical insulator, *PNAS* **9**, 111 (36) (2014).
- [39] P. Lo, C. D. Santangelo, B. G. Chen, C. Jian, K. Roychowdhury, and M. J. Lawler, Topology in Nonlinear Mechanical Systems, *Phys. Rev. Lett.* **127**, 076802 (2021).
- [40] L. Jezequel, P. Delplace, Nonlinear edge modes from topological 1D lattices, arXiv:2107.10016.
- [41] J. R. Tempelman, K. H. Matlack, and A. F. Vakakis, Topological Protection in a Strongly Nonlinear Interface Lattice, arXiv:2105.08137.
- [42] C. L. Kane and T. C. Lubensky, Topological boundary modes in isostatic lattices, *Nat. Phys.* **10** (2014).
- [43] M-Z Liu, C-Q Cao, X-Q Zhu, B-N Liu, and K-C Peng, Variational Principles and Solitary Wave Solutions of Generalized Nonlinear Schrödinger Equation in the Ocean, *Journal of Applied and Computational Mechanics*, Articles in Press (2021).
- [44] M. Ezawa, Topological Toda lattice and nonlinear bulk-edge correspondence, arXiv:2105.10851.
- [45] M. Ezawa, Nonlinear topological phase diagram in dimerized sine-Gordon model, arXiv:2110.15602v1.

REVIEWER COMMENTS

Reviewer #1 (Remarks to the Author):

After reading the revised manuscript, it is still very hard to read this paper. Although I have consulted some literatures on nonlinear topological physics, it is a hard task to judge whether this formalism is valid or not. It will be almost impossible to understand the contents even for specialists. In this context, this paper should be submitted to a more specialized journal such as mathematics oriented journals.

There are still some concerns on the manuscript before transferring the manuscript.

1)

The authors use a plane-wave ansatz (2) even for strong nonlinear systems. However, the momentum is not a good number in the nonlinear system as is clear from the fact that nonlinear Schrodinger term is not momentum diagonal. Hence, it is impossible to use a momentum to identify the system for strong nonlinear systems. The authors should clarify on this point.

2)

The same critique is true also for Fig.2, where the band structure is plotted as a function of "q". It is impossible to plot a band structure as a function of momentum in the nonlinear system.

3)

The authors should apply the formalism to well-known models such as the ordinary nonlinear SSH model and confirm that the results are valid.

4)

Is it possible to find nonlinearity induced topological phase transition as shown in "Nonlinearity-induced photonic topological insulator"

Science 6 Nov 2020 Vol 370, Issue 6517 pp. 701-704 DOI: 10.1126/science.abd2033

Reviewer #2 (Remarks to the Author):

In this revision, the authors have addressed my two main concerns by providing new arguments and calculations that appear convincing to me. The responses to the questions of the author referees also seem reasonably satisfactory to me. Therefore, I recommend publication of this version in Nature Communications.

Reply to Reviewer 1

Reviewer's remarks: *After reading the revised manuscript, it is still very hard to read this paper. Although I have consulted some literatures on nonlinear topological physics, it is a hard task to judge whether this formalism is valid or not. It will be almost impossible to understand the contents even for specialists. In this context, this paper should be submitted to a more specialized journal such as mathematics-oriented journals. There are still some concerns on the manuscript before transferring the manuscript.*

Our Reply: We thank the reviewer for consulting related literatures and raising these concerns, which lie in three aspects:

(1) Currently, there are few strong nonlinear topological literatures to judge our formalism. The lack of previous study rightly demonstrates the cutting edge of strong nonlinear topology and the innovation of our manuscript, which timely fills the blank of general-nonlinear dynamics and topological physics.

(2) Our work appears mathematical, that is because our work combines nonlinear dynamics --- an area of math community, and topology --- an abstract notion originated from mathematicians. Both nonlinearity and topology have become the important branches and frontier of physics. Our work indeed bridges the gap between topological physics and general-nonlinear dynamics. We try our best to improve the readability by adding intermediate steps of derivations (e.g. Eqs.SI.(1.10), (2.17), and (6.5)), additional explanations (e.g., Eq.SI.(1.12) of supplementary information, and the paragraph that follows), and physical pictures (e.g., main text Page.2, right column).

(3) As highlighted by papers of electrical and photonic metamaterials (e.g. *Nat. Electron* **1**, 178 (2018), *Science* **370**, 701 (2020)), we believe the publication of our manuscript on *Nat. Commun.* will soon enable experiments and applications of the strong nonlinear topological system. Furthermore, the publication will cause the general interests of broad audience from the communities of nonlinear physics (e.g. *Phys. Rev. Lett.* **125**, 258301 (2020)),

electromagnetic (e.g. *Phys. Rev. Lett.* **100**, 063901 (2008)) and photonic metamaterials (e.g. *Nat. Commun.* **7**, 1 (2016)), and biophysics (*PNAS* **115** E9031 (2018)), etc.

Thus, we respectfully state that our work has the importance and generality to reach the high standard of Nature Communications.

Reviewer comments: 1) *The authors use a plane-wave ansatz (2) even for strong nonlinear systems. However, the momentum is not a good number in the nonlinear system as is clear from the fact that nonlinear Schrodinger term is not momentum diagonal. Hence, it is impossible to use a momentum to identify the system for strong nonlinear systems. The authors should clarify on this point.*

Fig.R1: **Linear** plane waves are e^{iqx} (red). The waveform of nonlinear normal modes are not e^{iqx} (blue, supplementary information, Page.24, Fig.SI.11(a)).

Our Reply: We thank the reviewer for bringing this question. We kindly remind that the reviewer may have neglected that wavenumber q is a good number and has been widely used to identify **plane-wave nonlinear normal modes** in textbook [2], Wikipedia [3], and articles [4-7].

In fact, for **translationally invariant nonlinear interactions**, plane-wave nonlinear normal modes are well-defined periodic oscillations, and q is a good number to identify their phase shift (see textbook of Ref. [2] and Wikipedia in Ref. [3]). As shown in Fig. R1, unlike the waveform e^{iqx} for **linear** plane waves, **plane-wave nonlinear normal modes are not e^{iqx}** . For example, strong

nonlinear plane waves of classical quartic Klein-Gordon equations are Jacobi Elliptic functions [3], where q characterizes the phase shift. Jacobi-Elliptic waves are also studied in gravitational waves for classical Minkowskian Yang-Mills theory [4], where q is the wavenumber. Nonlinear plane waves ubiquitously arise in compressible atmosphere [5], porous media [6], and mechanical lattices [7], where q is the wavenumber. In summary, the wavenumber q is a good number to identify the site-to-site phase shift of plane-wave nonlinear normal modes.

Summary of the modifications: We add the aforementioned references in the revised main text, Page.2, right column, Paragraph.2: “It takes the plane-wave nonlinear normal modes in translationally invariant systems [31, 59–63] (also dubbed as “nonlinear plane waves”)”. In Page.2, right column, Paragraph.3, we add a new sentence marked by blue that refers the ansatz to these literatures: “First, existing works [60] manifest this form of nonlinear plane waves, such as classical Minkowskian Yang-Mills theory [61], compressible atmosphere [62], porous media [63], and mechanical lattices [32,55].” In the revised supplementary information, Page.1, left column, Paragraph.3 marked by blue, we address a more detailed discussion on these literatures.

Reviewer comments: *2) The same critique is true also for Fig.2, where the band structure is plotted as a function of “ q ”. It is impossible to plot a band structure as a function of momentum in the nonlinear system.*

Our Reply: As we mention above, wavenumber q is a good number for nonlinear plane waves in translationally invariant nonlinear systems [2-7]. Hence, the frequencies of temporal-periodic nonlinear plane waves are characterized by the wavenumber q .

Summary of the modifications: In the updated main text, we add a new sentence in Page.2, right column, Paragraph.3: “Consequently, the frequencies

and band structure of temporal-periodic nonlinear bulk waves [32,55,60-63] are characterized by the wave number q as well, as pictorially indicated in Fig.1(e).”

Reviewer comments: 3) *The authors should apply the formalism to well-known models such as the ordinary nonlinear SSH model and confirm that the results are valid.*

Our reply: We thank the reviewer for raising this point. We have discussed the ordinary nonlinear SSH model in Sec. VI of the supplementary information. Our work can indeed address the topological properties of this model.

Fig.R2: The results of ordinary nonlinear SSH model derived from our general formalism. (a) Band structure derived by Eq.SI.(6.3), (b) and (c) The frequency profile and spatial profile of topological edge modes derived by Eq.SI.(6.10).

Our general-nonlinear topological index (Eq.(3) of the main text) can reduce to the winding number of ordinary nonlinear SSH models [9], as shown in Eq.SI.(6.5). General-nonlinear dispersion relation reduce to that of ordinary nonlinear SSH model, as pictorially depicted by Fig.R2 (a) (see Fig.SI.13(a)). General-nonlinear topological edge modes also reduce to the edge modes of ordinary nonlinear SSH model, as depicted by Figs.R2 (b, c) (see Figs.SI.13(d, f)).

Reviewer comments 4) *Is it possible to find nonlinearity induced topological phase transition as shown in “Nonlinearity-induced photonic topological*

insulator" Science 6 Nov 2020 Vol 370, Issue 6517 pp. 701-704 DOI: 10.1126/science.abd203.

Our Reply: We appreciate the reviewer for raising the paper of Ref. [1], Science **370** 701 (2020). Our formalism can indeed find the nonlinearity induced topological phase transition, because we can derive the Berry connection of Ref. [1].

Ref. [1] studies the topological phases in a (2+1)-D photonic metamaterial, where the topological number is **3-form** Chern-Simons theory. Our work conducts 1D beyond-Kerr nonlinear models, where the topological number is the **1-form** Chern-Simons theory known as Berry phase. Despite the distinct physical origins, the topological invariants are both constructed from Berry connection [10]. Ref. [1] constructs Berry connection under Kerr-nonlinear interactions, whereas our formalism establishes Berry connection for general nonlinearities. Using Kerr-nonlinearity, our formalism naturally derives the Berry connection of Ref. [1], and finds Kerr-nonlinearity-induced topological transition [1]. The subject of this work is to focus on 1D general-nonlinear topological systems. In our following research, we will conduct the topological properties of 3D general-nonlinear models.

Summary of the modifications: We thank the reviewer for raising this question. We add the discussions of Kerr-nonlinearity induced topological phase transition [1] in the supplementary information, Sec. I, Page.2, right column, Paragraph.2 marked by blue. We address the physical differences between Ref. [1] and our work, and discuss the relationship between Kerr and general-nonlinear Berry connections. Meanwhile, we add a new sentence in the main text, Page.8, left column, Paragraph.3, marked by blue, to discuss the future direction of general-nonlinear 3D topological insulators.

Reply to Reviewer 2

Reviewer's remarks: *In this revision, the authors have addressed my two main*

concerns by providing new arguments and calculations that appear convincing to me. The responses to the questions of the author referees also seem reasonably satisfactory to me. Therefore, I recommend publication of this version in Nature Communications.

Our Reply: We really appreciate your strong recommendation on the publication of our manuscript. Our work benefits quite a lot from the discussions with your insightful referee questions.

With best regards,

Di Zhou, D. Zeb Rocklin, Michael J. Leamy, and Yugui Yao

References

- [1] L. J. Maczewsky, M. Heinrich, M. Kremer, S. K. Ivanov, M. Ehrhardt, F. Martinez, Y. V. Kartashov, V. V. Konotop, L. Torner, D. Bauer, and A. Szameit, Nonlinearity-induced photonic topological insulator, *Science* **370**, 701 (2020).
- [2] A. F. Vakakis, L. I. Manevitch, Y. V. Mikhlin, V. N. Pilipchuk, and A. A. Zevin, *Normal modes and localization in nonlinear systems* (Springer, 2001); J. R. Tempelman, K. H. Matlack, and A. F. Vakakis, Topological protection in a strongly nonlinear interface lattice, *Phys. Rev. B* **104**, 174306 (2021).
- [3] https://en.wikipedia.org/wiki/Quartic_interaction; M. Frasca, Exact solutions of classical scalar field equations, *Journal of Nonlinear Mathematical Physics*. **18** (2): 291–297 (2011).
- [4] A. Tsapalis, E. P. Politis, X. N. Maintas, and F. K. Diakonov, Gauss' law and nonlinear plane waves for Yang-Mills theory, *Phys. Rev. D* **93**, 085003 (2016).
- [5] M. Schlutow and E. Wahlen, Generalized modulation theory for strongly nonlinear gravity waves in a compressible atmosphere, *Mathematics of Climate and Weather Forecasting*, **6**, 97 (2020).
- [6] H. Benjamin, Nonlinear plane waves in saturated porous media with incompressible constituents, *Proc. R. Soc. A* **477**: 20210086 (2021).

- [7] M. D. Fronk and M. J. Leamy, Higher-Order Dispersion, Stability, and Waveform Invariance in Nonlinear Monoatomic and Diatomic Systems, *Journal of Vibration and Acoustics*, **139**, 051003 (2017).
- [8] E. Fradkin, *Field theories of condensed matter physics*, Cambridge University Press, (2013).
- [9] Y. Hadad, A. B. Khanikaev, and A. Alu, Self-induced topological transitions and edge states supported by nonlinear staggered potentials, *Phys. Rev. B* **93**, 155112 (2016)
- [10] M. Z. Hasan and J. E. Moore, Three-Dimensional Topological Insulators, *Annu. Rev. Condens. Matter Phys.* 2011. 2:55–78.

REVIEWERS' COMMENTS

Reviewer #1 (Remarks to the Author):

The authors do not address my previous comments seriously.
I cannot find any reason to continue the review process.
Could you please send it to additional referees, if necessary?

Reviewer #2 (Remarks to the Author):

I acknowledge the effort of the authors to provide detailed answers to all the points raised by me and the other Referees. Considering all the material, I am happy to recommend publication of the manuscript in Nature Communications.

Reply to Referee 1

Referee's remarks: *The authors do not address my previous comments seriously. I cannot find any reason to continue the review process. Could you please send it to additional referees, if necessary?*

Our Reply: We thank the referee for the careful review of our manuscript and the reply. In fact, as Referee #2 points out, we have carefully addressed the referee comments of the last round. By answering the referee comments, our manuscript significantly improves on clarifying the notions of nonlinear dynamics, such as nonlinear normal modes and nonlinear band structures, based on the references of existing literatures.

Reply to Reviewer 2

Reviewer's remarks: *I acknowledge the effort of the authors to provide detailed answers to all the points raised by me and the other Referees. Considering all the material, I am happy to recommend publication of the manuscript in Nature Communications.*

Our Reply: We really appreciate your strong recommendation on the publication of our manuscript. Our work improves quite a lot from the discussions with your insightful reviewer questions.

With best regards,

Di Zhou, D. Zeb Rocklin, Michael J. Leamy, and Yugui Yao